# INFERENCE OF EVOLVING MENTAL STATES FROM IRREGULAR ACTION EVENTS TO UNDERSTAND HUMAN BEHAVIORS

## ABSTRACT

Inference of *latent human mental processes*, such as *belief*, *intention*, or *desire*, is crucial for developing AI with human-like intelligence, enabling more effective and timely collaboration. In this paper, we introduce a versatile encoder-decoder model designed to infer **evolving mental processes** based on **irregularly observed action events** and predict future occurrences. The primary challenges arise from two factors: both actions and mental processes are irregular events, and the observed action data is often limited. To address the irregularity of these events, we leverage a **temporal point process model** within the encoder-decoder framework, effectively capturing the dynamics of both action and mental events. Additionally, we implement a *backtracking mechanism* in the decoder to enhance the accuracy of predicting future actions and evolving mental states. To tackle the issue of limited data, our model incorporates **logic rules** as priors, enabling accurate inferences from just a few observed samples. These logic rules can be refined and updated as needed, providing flexibility to the model. Overall, our approach enhances the understanding of human behavior by predicting when actions will occur and how mental processes evolve. Experiments on both synthetic and real-world datasets demonstrate the strong performance of our model in inferring mental states and predicting future actions, contributing to the development of more human-centric AI systems.

## 1 INTRODUCTION

In the rapidly advancing field of artificial intelligence, there is a growing interest in the development of autonomous agents capable of collaborating with humans across various tasks (Carroll et al., 2019; Puig et al., 2020; Strouse et al., 2021). Effective collaboration relies on these agents' capacity to **understand human actions and accurately infer underlying intentions**, enabling timely and appropriate assistance.

Human daily activities consist of intricate sequences of actions, where the successful execution of complex tasks is contingent upon the precise ordering of these actions, driven by specific intentions. For instance (as illustrated in Fig. 1), when preparing oatmeal, an individual typically follows a discernible sequence of actions based on intentions: entering the kitchen, retrieving a cup, taking oatmeal, pouring it, and subsequently cleaning the table (Damen et al., 2018). This example underscores how intentions can guide actions, illustrating the need for AI systems to comprehend the underlying intentions of human behavior.

Although prior research has focused on forecasting future actions based on observed sequences (Abu Farha et al., 2018; Cramer et al., 2021; Darvish et al., 2020; Furnari & Farinella, 2020), the inherent irregularity of these actions poses significant challenges. The time intervals between actions convey crucial information regarding the underlying intentions and dynamics of human behavior; neglecting these intervals may result in AI systems overlooking vital contextual cues. Furthermore, differing intentions can lead to deviations in subsequent actions, complicating AI agents' ability to fully comprehend human behavior and accurately predict future actions (Hu & Clune, 2023; Roy & Fernando, 2022; Zolotas & Demiris, 2022).

It is noteworthy that while human behaviors may appear complex, the **underlying logic governing these actions and mental states is often straightforward, clear, and generalizable** (Northrop, 1947). Logic rules serve as compact representations of knowledge that delineate likely actions based on specific conditions. By incorporating intuitive logic rules as prior knowledge, AI agents can significantly enhance their capacity to infer human mental states, predict forthcoming actions and tackle with limited data challenges.

In this paper, we propose a novel model based on a well-structured **encoder-decoder architecture** designed to infer unobserved human intentions at fine-grained time resolutions from continuously observed irregular action events. To tackle the challenges of irregularity, our encoder utilizes a self-attention mechanism to map these irregular actions and their timestamps onto a discretized timeline. The latent mental state is modeled as a discrete-time renewal process, with the encoder using action sequence embeddings to estimate occurrence probabilities for each grid. This inference process dynamically integrates historical and future information, facilitating a comprehensive representation of hidden mental state evolution.

Upon obtaining the inferred mental states, we employ a rule-informed decoder to generate actions using the **temporal point process** (TPP) models. The logic rules can be predefined or refined through our model, which incorporates a **rule generator** utilizing an efficient column generation algorithm (Barnhart et al., 1998; Li et al., 2021) to uncover latent rules. This generator mitigates inaccuracies associated with hand-crafted knowledge and facilitates the exploration of potentially overlooked rules. The decoder utilizes the mined rules, along with observed actions and sampled mental events, to derive conditional intensity functions that enable action generation.

The overall learning objective function is grounded in the variational lower bound, with the encoder-decoder and rule generator trained jointly. To enhance the accuracy of real-time action predictions, we introduce a novel backtracking action sampling mechanism. This mechanism iteratively refines predictions by revisiting and adjusting previously inferred mental states in response to newly observed actions and contextual information, thereby increasing the model's adaptability to real-time fluctuations in human thoughts and behaviors.

**Contributions:** The contributions of this work are three-folds: *i)* We introduce a well-designed encoder-decoder architecture that infers unobserved human intentions in fine-grained time resolutions, effectively addressing irregular action events. *ii)* The model employs a rule generator that uncovers latent rules, enhancing adaptability and accuracy by integrating both predefined and refined logic rules, tackling with limited data. *iii)* A novel backtracking action sampling mechanism iteratively refines predictions, improving responsiveness to real-time fluctuations in human behavior.

## 2 RELATED WORK

**Neural Temporal Point Process (TPP)**  TPP provides an elegant model for irregular events in continuous time, which are characterized by the event intensity function. Over the past decades, researchers have focused on enhancing the flexibility of this intensity function. Some approaches utilize recurrent neural networks (RNNs) and long short-term memory (LSTM) networks, such as (Du et al., 2016; Mei & Eisner, 2017; Xiao et al., 2017; Omi et al., 2019; Shchur et al., 2019; Mei et al., 2020; Boyd et al., 2020). Others leverage Transformer architectures (Vaswani et al., 2017), including (Zuo et al., 2020; Zhang et al., 2020; Enguehard et al., 2020; Sharma et al., 2021; Zhu et al., 2021; Yang et al., 2021). Recently, Xue et al. (2022) combined the base TPP with an energy function to overcome the cascading errors of auto-regressive models in making predictions, *which is also what our model aims to avoid*. The work by Lüdke et al. (2023) employed a probabilistic denoising diffusion model for TPP, addressing similar issues. *In our framework, we leverage TPP to model the interleaved observed action events and latent mental events, aiding the design of the encoder and decoder.*

**Enhance Interpretability via Logic Rules**  Logic rules effectively represent domain knowledge and hypotheses, offering explanations for real-world event data. Previous studies have utilized predefined logic rules as prior knowledge to enhance model performance. For example, (Liu et al., 2023) formalized traffic laws using linear temporal logic derived from government regulations and expert insights. Similarly, Li et al. (2020) incorporated first-order logic rules, summarized by human experts, to model event dynamics via intensity functions. Other works, such as (Zhang et al., 2021), has also employed temporal logic encoded prior knowledge. Recently, some research has focused on

learning rules from data. For instance, Li et al. (2021) introduced a temporal rule learning algorithm based on column generation, while Yan et al. (2023) developed a differentiable neuro-symbolic framework for modeling TPPs by learning weighted clock logic formulas. Cao et al. (2024) presents a model for learning spatial-temporal logic rules to explain human actions that are closely related to ours; however, it does not infer fine-grained latent mental events in real time. *Our model strikes a balance by leveraging rules as prior knowledge while maintaining the ability to refine these rules.*

**Latent Variable Inference** Latent variable inference presents a fundamental challenge. The Expectation-Maximization (EM) algorithm (Dempster et al., 1977) provides a robust iterative method for learning latent variable models, particularly when posterior distributions are tractable. In the context of human intentions as latent variables, Wei et al. (2017) developed an EM-based approach for their inference. Additionally, spectral algorithms (Hsu et al., 2012; Kulesza et al., 2014) are recognized for their computational efficiency and provable guarantees, enhancing the feasibility of inference tasks. Variational inference methods (Kingma & Welling, 2013) and wake-sleep algorithms (Bornschein & Bengio, 2014) are commonly employed to approximate intractable posterior distributions. The Variational Autoencoder (VAE) framework (Kingma & Welling, 2013) encodes input data into latent variables, subsequently reconstructing the data from these latent representations. Notably, Mehrasa et al. (2019) leveraged the VAE architecture to synthesize human trajectories, albeit without addressing the interpretability of the latent variables. In a similar vein, Zolotas & Demiris (2022) explored VAE-based models for inferring human intentions but were constrained to discrete intent variables. Hidden Markov Models (HMMs) have also been employed for modeling sequential data and inferring hidden states. For instance, Foti et al. (2014) proposed a stochastic variational inference algorithm for estimating HMM parameters, while Jeong et al. (2021) integrated HMMs with VAEs to infer human activity sequences. *In contrast, our model aim to infer the evolving latent mental processes in continuous time when the observed action events are irregular.*

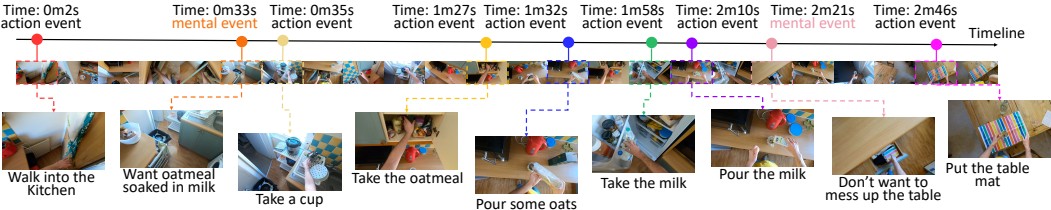

Figure 1: An illustrative example of preparing oatmeal with milk, showing actions are driven by intentions. At 0m33s, the individual formed the intention to have oatmeal with milk, and at 2m21s, one intended to keep the table clean. He subsequently took appropriate actions, such as grabbing a cup and pouring milk.

## 3 PRELIMINARIES

We assume both actions and mental processes are temporal point processes (TPPs). To address the sporadic nature of mental processes and facilitate subsequent inference, we represent them as discrete-time renewal processes (DT-RPs). It enables us to approximate the intensity function of the latent mental process by stitching together conditional hazard functions. Furthermore, the discrete conditional hazard rate, can be computed using instantaneous event probability and survival rate functions. As the discrete interval approaches infinitesimal, it approximates the continuous conditional hazard rate function, thus approximating the continuous conditional intensity function of the mental processes. Building on this foundation, we will infer latent mental states in discrete time using a DT-RP encoder, while action events will be generated by the decoder in continuous time, modeled by TPPs.

**Notation** Define an action set $\mathcal{A}$ that produces a fully observed action sequence $\boldsymbol{a}_{1:N_1} = \left\{(t_i^a, k_i^a)\right\}_{i=1}^{N_1}$ on $[0, t)$, where $t_i^a$ is the occurrence time and $k_i^a \in \mathcal{A}$ is the event type. Similarly, a latent mental sequence $\boldsymbol{m}_{1:N_2} = \left\{(t_j^m, k_j^m)\right\}_{j=1}^{N_2}$ is defined for mental events, with $k_j^m \in \mathcal{M}$. The set of all events is $\mathcal{X} = \mathcal{A} \cup \mathcal{M}$, and the history up to time $t$ is denoted as $\mathcal{H}_t = \boldsymbol{a} \cup \boldsymbol{m}$.

**Discrete-Time Renewal Process** Consider an example, *someone who is thinking about starting an exercise routine. Each time they delay the decision, their mental state resets, and the time until the next urge to exercise follows this survival process. Initially, the chance of feeling the urge is high, but as time passes without taking action, this likelihood decreases. When a new mental*

*trigger arises, the process resets, illustrating the cyclical nature of decision-making.* Therefore, we model the mental processes as discrete-time renewal processes, fundamentally akin to temporal point processes. After each event, the system resets, and the time until the next event follows a survival process, where the survival probability starts at 1 and gradually decreases.

We first discretize the time horizon into intervals $V_\xi = (t_{\xi-1}, t_\xi]$, with $0 < t_1 < t_2 < \cdots < t_L < T$. The intensity function can be estimating by stitching together conditional hazard functions (Rasmussen, 2018), which captures the evolution of events based on the elapsed time since the last event and defined as the conditional probability as:

$$h_\xi = \mathbb{P}(t_j^m \in V_\xi | t_j^m > t_{\xi-1}) = p_\xi / S(t_{\xi-1})$$

The event and survival rate functions for the mental events over discrete time space are:

$$W(t_\xi) = \mathbb{P}(t_j^m \le t_\xi) = \sum_{t_j^m \le t_\xi} \mathbb{P}(t_j^m \in V_\xi), \quad S(t_\xi) = \mathbb{P}(t_j^m > t_\xi) = \sum_{t_j^m > t_\xi} \mathbb{P}(t_j^m \in V_\xi). \tag{1}$$

And the discrete mental event time probability function at the $\xi$-th time interval is

$$p_\xi = \mathbb{P}(t_j^m \in V_\xi) = W(t_\xi) - W(t_{\xi-1}) = S(t_{\xi-1}) - S(t_\xi) \tag{2}$$

which not only approximates the continuous probability function as the intervals $V_\xi$ approach infinitesimal, but also enables the derivation of the hazard function, providing an approximation of the intensity of the mental process as we assume. After an event, $S(t_\xi)$ resets to 1 at the next interval, ensuring that both the survival and event probabilities reflect the reset process.

**Temporal Logic Point Process (TLPP)** To leverage prior knowledge to guide the inference and tackle limited-data challange, we use temporal logic rules (Li et al., 2020; 2021) to construct intensity functions for action events. Let $x^u$ be a grounded Boolean logic variable, which is true (i.e., 1) at the occurrence time and is false (i.e., 0) otherwise. A general temporal logic rule is given by:

$$f : y \leftarrow \bigwedge_{x^u \in \mathcal{X}_f} x^u \bigwedge_{x^u, x^v \in \mathcal{X}_f} R_j(x^u, x^v)$$

where $R_j(\cdot)$ defines temporal relations such as "Before", "Equal", or "After", which can be grounded by the event times. Given the rules and the historical data, the intensity function, denoted as $\lambda_y(t|\mathcal{H}_t)$ for the head predicate $y$, is computed as:

$$\lambda_y(t|\mathcal{H}_t) = \mu_y + \sum_{f \in \mathcal{F}} w_f \phi_f(\mathcal{H}_t),$$

where $\mu_y$ is the base intensity, the term $\phi_f(\mathcal{H}_t)$ captures logic-informed features from historical data, accounting for effective combinations of historical events that satisfy the body condition of the rule. When this condition is met, the rule is triggered, and the count of combinations that make the Boolean body condition true is included to form the feature. Here, $\mathcal{F}$ denotes the set of rules, and $w_f$ represents the weights assigned to each rule.

Given intensity function, the likelihood function of event data observed in $[0, T)$ is computed as:

$$\mathcal{L} = \prod_{y \in \mathcal{X}} \left( \prod_{i=1}^{N_y} \lambda_y(t_i|\mathcal{H}_{t_i}) \exp\left( -\int_0^T \lambda_y(\tau|\mathcal{H}_\tau)d\tau \right) \right), \tag{3}$$

where $N_y$ is the total count of event $y$.

## 4 METHOD

We propose a flexible model for asynchronous action processes modeling as well as inferring latent mental processes. The architecture of our model is illustrated in Fig. 2. Overall, our model employs an encoder-decoder architecture that incorporates a rule generator which use predefined temporal logic rule templates as prior knowledge to generate rules via column generation. Additionally, we introduce a backtracking mechanism to ensure stable prediction of future events.

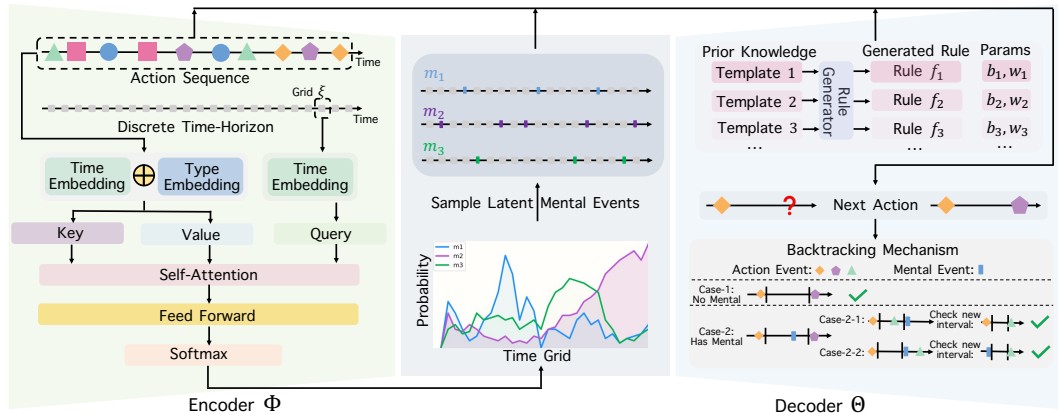

Figure 2: Model architecture: A encoder-decoder framework that incorporates a rule generator.

## 4.1 ENCODER

**Action Sequence Embedding** To capture the information of action sequence on the spanning time horizon, we resort to the self-attention mechanism to encode the observed irregular action events, which will be used to model the occurrence probability of the mental events in each discrete time interval. Given action events, we first compute time embedding for the observed action time

$$[\boldsymbol{z}(t_i^a)]_d = \begin{cases} \cos(t_i^a/10000^{\frac{d-1}{D}}), & \text{if } d \text{ is odd} \\ \sin(t_i^a/10000^{\frac{d}{D}}), & \text{if } d \text{ is even} \end{cases} \tag{4}$$

for each action occurrence time $t_i^a$, we deterministically compute $\boldsymbol{z}(t_i^a) \in \mathbb{R}^D$, where $D$ is the dimension of embedding. We also train an embedding matrix $\boldsymbol{U} \in \mathbb{R}^{D \times |\mathcal{A}|}$ for action event type, where the $k$-th column of $\boldsymbol{U}$ is a $D$-dimensional embedding for action event type $k^a$. For any action event type $k_i^a$, we denote the corresponding one-hot encoding as $\boldsymbol{k}_i$. Then, we represent the type embedding of this action event as $\boldsymbol{U}\boldsymbol{k}_i \in \mathbb{R}^D$. Then the embedding of the action event sequence $\boldsymbol{a}_{1:N_1} = \{(t_i^a, k_i^a)\}_{i=1}^{N_1}$ is then specified by $\boldsymbol{X} = (\boldsymbol{U}\boldsymbol{Y} + \boldsymbol{Z})^\top$, where $\boldsymbol{Y} = [\boldsymbol{k}_1, \boldsymbol{k}_2, ..., \boldsymbol{k}_{N_1}] \in \mathbb{R}^{|\mathcal{A}| \times N_1}$ is the collection of action event type embedding, and $\boldsymbol{Z} = [\boldsymbol{z}(t_1^a), \boldsymbol{z}(t_2^a), ..., \boldsymbol{z}(t_{N_1}^a)] \in \mathbb{R}^{D \times N_1}$ is the concatenation of action event time embeddings. $\boldsymbol{X} \in \mathbb{R}^{N_1 \times D}$ and each row of $\boldsymbol{X}$ corresponds to the embedding of a specific action event in the sequence.

As discussed in Sec. 3, when we infer the latent mental state, time horizon is divided into disjoint intervals $V_\xi = (t_{\xi-1}, t_\xi]$, where $\xi = 1, 2, ..., L$. We will use the action sequence embedding to model the probability of mental events. We apply the same trigonometric function time embedding as Eq. (4) on the disjoint small intervals. For simplicity, we represent these intervals using their upper bound $t_\xi$. Then the embedding of the discrete timeline is given by $\boldsymbol{L} = [\boldsymbol{z}(t_1), \boldsymbol{z}(t_2), ..., \boldsymbol{z}(t_\xi), ..., \boldsymbol{z}(t_L)] \in \mathbb{R}^{D \times L}$.

After the initial embedding layers, we pass $\boldsymbol{X}$ and $\boldsymbol{L}$ through the self-attention module. Specifically, we compute the attention output $\boldsymbol{S}$ by

$$\boldsymbol{Q} = \boldsymbol{L}\boldsymbol{W}^Q, \boldsymbol{K} = \boldsymbol{X}\boldsymbol{W}^K, \boldsymbol{V} = \boldsymbol{X}\boldsymbol{W}^V, \boldsymbol{S} = \text{Softmax}\left(\frac{\boldsymbol{Q}\boldsymbol{K}^\top}{\sqrt{D_K}}\right)\boldsymbol{V} \tag{5}$$

where $\boldsymbol{Q}, \boldsymbol{K}$, and $\boldsymbol{V}$ are the query, key, and value matrices. The query matrix is obtained by some transformation on the embedding of the discrete timeline $\boldsymbol{L}$, while key and value matrices are obtained by different transformations of $\boldsymbol{X}$. $\boldsymbol{W}^Q \in \mathbb{R}^{D \times D_Q}, \boldsymbol{W}^K \in \mathbb{R}^{D \times D_K}$, and $\boldsymbol{W}^V \in \mathbb{R}^{D \times D_V}$ are weights for the linear transformations respectively. We also incorporate multi-head self-attention to enhance the flexibility of the model, thereby yielding greater advantages in data fitting (Vaswani et al., 2017; Zuo et al., 2020). We highlight here that by doing so, each disjoint small interval can capture the information of entire action sequence within the time horizon, which enables the subsequent probability distribution modeling for latent mental event occurrences.

**Infer Mental Probability** Next, we feed the attention output through a position-wise feed-forward neural network to model the probability distribution of latent mental event occurrences. Consider

$S_\xi \in \mathbb{R}^D$ is the $\xi$-th row of the attention output, we update the hidden vector after receiving the current input,

$$h_\xi = h(S_\xi) \tag{6}$$

where $h(\cdot)$ represents the neural network. We map each hidden vector to the probability of any mental event at the $V_\xi$-th time interval

$$p_\xi = \text{Softmax}(W^P h_\xi^\top) \tag{7}$$

where $p_\xi \in \mathbb{R}^{|\mathcal{M}|+1}$ is the probability of any mental event at the $V_\xi$-th time interval (as defined in Eq. (2)) and $W^P \in \mathbb{R}^{(|\mathcal{M}|+1)\times D}$ is the weight matrix for mental event occurrence probability. $(|\mathcal{M}|+1)$ represents the total number of mental event types. Here we consider the circumstance where no mental event can occur within the interval, and thus we add 1 to the dimension.

Then we can sample latent mental event according to the computed probability $p_\xi$ at each discrete time interval. To enable the computation of gradients, we leverage the re-parametrization trick (Jang et al., 2016). Let $g_\xi \sim \text{Gumbel}(0,1)$, we can sample the mental event type at time interval $V_\xi$,

$$k_\xi^{\tilde{m}} \sim \text{Softmax}\left(\frac{1}{\alpha}\left(\log(p_\xi^1) + g_\xi^1, ..., \log(p_\xi^{|\mathcal{M}|+1}) + g_\xi^{|\mathcal{M}|+1}\right)\right) \tag{8}$$

where $\alpha$ is the temperature parameter. The sampled mental event time $t_\xi^{\tilde{m}}$ is the corresponding upper bound of each discrete time interval. Therefore, the sampled mental event at the $V_\xi$-th time interval is $\tilde{m}_\xi = (t_\xi^{\tilde{m}}, k_\xi^{\tilde{m}})$

We denote the fitted probability distribution of any mental event before current time $\xi$ as $q_{\Phi,\xi}(m) = [p_1, p_2, ..., p_\xi] \in \mathbb{R}^{(|\mathcal{M}|+1)\times\xi}$ and all the sampled latent mental event sequence as $\tilde{m}_{1:\xi} = \{\tilde{m}_1, \tilde{m}_2, ...., \tilde{m}_\xi\} \sim q_{\Phi,\xi}(m)$. Here, $\Phi$ represents all the parameters of the encoder.

## 4.2 DECODER

The decoder is designed based on the insight that the inferred mental event sequences as well as the observed historical actions will jointly influence the occurrence of future action events. Built upon the TLPP (as discussed in Sec. 3), the parameters of the decoder are the bases and weights of the generated logic rules, denoted as $\Theta = [\mu, w]$. The goal of decoder is to generate next action $\hat{a}_i = (t_i^{\hat{a}}, k_i^{\hat{a}})$ given a sequence of observed past actions $a_{1:i-1}$ and sampled latent mental events before the last action $a_{i-1}$, denoted as $\tilde{m}_{\lfloor a_{i-1} \rfloor}$, where $\lfloor \cdot \rfloor$ denotes the closest time interval before the last action $a_{i-1}$.

With reliable expert knowledge and well-derived logic rules, we can steer the reconstruction process. However, in scenarios where obtaining expert knowledge is challenging or extracting logic rules is difficult, we need a more flexible approach to acquire prior knowledge. Therefore, we introduce a plug-in rule generator that utilizes predefined logic templates to generate temporal logic rules, employing the column generation algorithm (Barnhart et al., 1998; Dash et al., 2018; Wei et al., 2019). In this framework, the rule generating problem is solved via two alternating procedures: the master-problem and the sub-problem (Li et al., 2021), where the master-problem aims to reweighting current rules, and the sub-problem is to search and construct a new temporal logic rule based on the given template, or generate rules from scratch (Please refer to Appendix. B for further details). In practice, we can employ two complementary strategies tailored to different problem settings: autonomous rule learning for data-rich domains and template-guided learning for scenarios with limited data but abundant prior knowledge. Every time we obtain the inferred latent mental events, the rule generator is trained on this data together with observed action events until stable logic rules emerge. These rules then guide the decoder's reconstruction process. Upon completion, rule bases and weights will update again synchronously with other model parameters.

Then we can compute the intensity for each action type $k^a$ that appear as head predicates in the learned logic rule set $\mathcal{F}$,

$$\lambda_{k^a}(t|a_{1:i-1}, \tilde{m}_{\lfloor a_{i-1}\rfloor}) = \mu_{k^a} + \sum_{f_{k^a}\in\mathcal{F}}\left(w_{f_{k^a}} \cdot \phi_{f_{k^a}}(a_{1:j-1}, \tilde{m}_{\lfloor a_{j-1}\rfloor})\right) \tag{9}$$

Denote current intensity for all action as

$$\lambda_a(t|a_{1:i-1}, \tilde{m}_{\lfloor a_{i-1}\rfloor}) = \sum_{k^a=1}^{|\mathcal{A}|} \lambda_{k^a}(t|a_{1:i-1}, \tilde{m}_{\lfloor a_{i-1}\rfloor}) \tag{10}$$

With logic-informed intensity function, next action time and type are estimated by,

$$p_{\boldsymbol{a}}(t|\boldsymbol{a}_{1:i-1}, \tilde{\boldsymbol{m}}_{\lfloor a_{i-1} \rfloor}) = \lambda_{\boldsymbol{a}}(t|\boldsymbol{a}_{1:i-1}, \tilde{\boldsymbol{m}}_{\lfloor a_{i-1} \rfloor}) \cdot \exp\left(-\int_{t_{i-1}^a}^{t} \lambda_{\boldsymbol{a}}(\tau|\mathcal{H}_\tau) d\tau\right) \quad (11)$$

$$\hat{t}_i^a = \int_{t_{i-1}^a}^{\infty} t \cdot p_{\boldsymbol{a}}(t|\boldsymbol{a}_{1:i-1}, \tilde{\boldsymbol{m}}_{\lfloor a_{i-1} \rfloor}) dt, \quad \hat{k}_i^a = \arg\max \frac{\lambda_{k^a}(t|\boldsymbol{a}_{1:i-1}, \tilde{\boldsymbol{m}}_{\lfloor a_{i-1} \rfloor})}{\lambda_{\boldsymbol{a}}(t|\boldsymbol{a}_{1:i-1}, \tilde{\boldsymbol{m}}_{\lfloor a_{i-1} \rfloor})} \quad (12)$$

where Eq. (11) is the likelihood that the next action will occur at time $t$ given the history.

## 4.3 LEARNING

As all the learnable parameters include $\boldsymbol{\Phi}$ and $\boldsymbol{\Theta}$, we consider variational lower bound (also known as ELBO) as our objective function, which can be represented as

$$\mathcal{L}(\boldsymbol{\Phi}, \boldsymbol{\Theta}) = \sum_{i=1}^{N_1} \left( \mathbb{E}_{\tilde{\boldsymbol{m}}_{\lfloor a_{i-1} \rfloor} \sim \boldsymbol{q}_{\boldsymbol{\Phi}, \lfloor a_{i-1} \rfloor}} \left[ \log \boldsymbol{p}_{\boldsymbol{\Theta}} \left( \boldsymbol{a}|\boldsymbol{a}_{1:i-1}, \tilde{\boldsymbol{m}}_{\lfloor a_{i-1} \rfloor} \right) - \log \boldsymbol{q}_{\boldsymbol{\Phi}, \lfloor a_{i-1} \rfloor} \left( \boldsymbol{m}|\boldsymbol{a}_{1:i-1} \right) \right] \right)$$
(13)

where $N_1$ is the total number of action events, $\boldsymbol{p}_{\boldsymbol{\Theta}}(\cdot)$ is the likelihood for next action, calculated as Eq. (11). $\boldsymbol{q}_{\boldsymbol{\Phi}, \lfloor a_{i-1} \rfloor}(\cdot)$ represents the fitted probability distribution for mental events before time $\lfloor a_{i-1} \rfloor$. We can evaluate the first term of Eq. (13) using Monte Carlo estimation by sampling mental events multiple times (Kingma & Welling, 2013).

$$\mathbb{E}_{\tilde{\boldsymbol{m}}_{\lfloor a_{i-1} \rfloor} \sim \boldsymbol{q}_{\boldsymbol{\Phi}, \lfloor a_{i-1} \rfloor}} \left[ \log \boldsymbol{p}_{\boldsymbol{\Theta}} \left( \boldsymbol{a}|\boldsymbol{a}_{1:i-1}, \tilde{\boldsymbol{m}}_{\lfloor a_{i-1} \rfloor} \right) \right] \simeq \frac{1}{S} \sum_{s=1}^{S} \log \boldsymbol{p}_{\boldsymbol{\Theta}} \left( \boldsymbol{a}|\boldsymbol{a}_{1:i-1}, \left( \tilde{\boldsymbol{m}}_{\lfloor a_{i-1} \rfloor} \right)_s \right). \quad (14)$$

For the second term of Eq. (13), we can represent it using conditional entropy

$$\mathbb{E}_{\tilde{\boldsymbol{m}}_{\lfloor a_{i-1} \rfloor} \sim \boldsymbol{q}_{\boldsymbol{\Phi}, \lfloor a_{i-1} \rfloor}} \left[ \log \boldsymbol{q}_{\boldsymbol{\Phi}, \lfloor a_{i-1} \rfloor} \left( \boldsymbol{m}|\boldsymbol{a}_{1:i-1} \right) \right] = H(\boldsymbol{m}|\boldsymbol{a}_{1:i-1}) = \sum_{\xi=1}^{\lfloor a_{i-1} \rfloor} H(m_\xi|m_{\xi-1}, ..., m_1, \boldsymbol{a}_{1:i-1})$$
(15)

where $H(m_\xi|m_{\xi-1}, ..., m_1, \boldsymbol{a}_{1:i-1}) = -\sum_{m=1}^{(|\mathcal{M}|+1)} p_\xi^m \log p_\xi^m$. And $p_\xi^m$ represents the occurrence probability for mental event type $m$ at small time interval $V_\xi$.

## 4.4 BACKTRACKING MECHANISM

In real-life scenarios, human behavior evolves with changing thoughts. These shifts in thinking are common and noteworthy, as humans often reflect on past thoughts before reaching final decisions. Based on this human cognitive and behavior logic, we need to prevent the omission of the crucial yet sparse mental events. Inspired by (Upadhyay et al., 2018), we propose a backtracking mechanism, which is detailed illustrated in Appendix. A, Alg. 1. If any mental event occurred during the time period between last action and the newly next action, first mental event within that time range should be considered, and the action will be *re-generated* accordingly, until no new mental event being adopted. By introducing this backtracking mechanism, our algorithm can consistently generate actions with relatively accurate samples of latent mental events. After the model is well-trained, this backtracking mechanism also plays a significant role in predicting future events.

## 5 EXPERIMENTS

### 5.1 EXPERIMENTAL SETUP

**Datasets** We conduct our experiments on both synthetic datasets and real-world datasets. While the results we report are based on small-scale dataset, our model is easily adopt to large-scale datasets. Details on scalability and computational cost analysis can be found in Appendix.H. For *synthetic datasets*, we simulate two datasets with same sample size (2000 sequences) and same time horizon (15s), but with different number of predicates and ground truth logic rules: *i) Syn Data-1*: 3 ground truth rules, 1 mental predicates and 2 action predicates. Each sequence has 18.60 actions on average, *ii) Syn Data-2*: a more complicated scenario with 4 ground truth rules, 2 mental predicates and 2 action predicates. Each sequence has 13.25 actions on average.

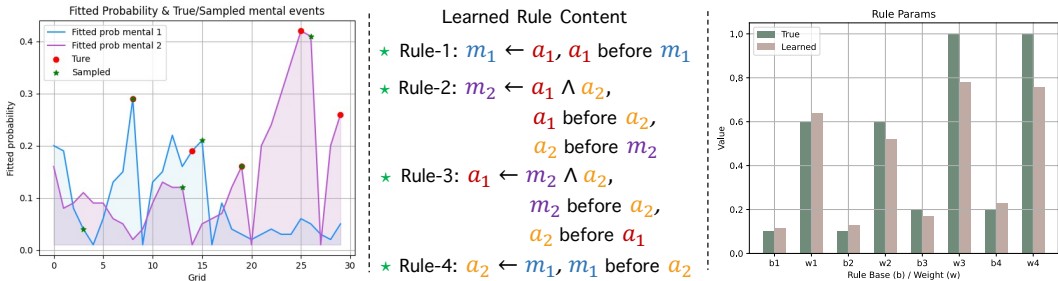

Figure 3: Results on Syn Data-2. **Left:** fitted occurrence probability of mental events for one sequence, corresponding sampled mental events, and ground truth mental events, **Middle:** learned rules given prior knowledge templates and green stars indicate that the rules are correctly learned. **Right:** ground truth rule parameters and learned parameters.

| Category | Model | Syn Data-1 | | Syn Data-2 | | Hand-Me-That | |
|---|---|---|---|---|---|---|---|
| | | ER%↓ | MAE↓ | ER%↓ | MAE↓ | ER%↓ | MAE↓ |
| Neural TPP | RMTPP | 48.37% | 3.11 | 52.14% | 3.20 | 79.86% | 2.13 |
| | THP | 45.46% | 2.83 | 48.93% | 2.99 | 78.85% | 2.08 |
| | PromptTPP | 43.47% | _2.42_ | _47.51%_ | 2.67 | 76.35% | _1.68_ |
| | HYPRO | 44.53% | 2.46 | 47.85% | _2.60_ | 75.88% | 1.70 |
| Rule-based Model | TELLER | 46.72% | 2.64 | 49.15% | 3.04 | 78.28% | 1.86 |
| | CLNN | 46.25% | 2.57 | 48.32% | 2.82 | 77.74% | 1.82 |
| | STLR | 45.05% | 2.52 | 48.23% | 2.72 | 77.25% | 1.80 |
| Gen. Model | AVAE | 45.13% | 2.82 | 47.53% | 2.92 | 80.12% | 2.10 |
| | GNTPP | 47.22% | 2.97 | 51.86% | 3.19 | 85.38% | 2.69 |
| | VEPP | 47.58% | 3.01 | 52.02% | 3.22 | 83.32% | 2.51 |
| | STVAE | 46.81% | 2.76 | 49.27% | 3.02 | 79.12% | 2.17 |
| – | **Ours*** | **41.72%** | **2.32** | **46.85%** | **2.52** | **75.28%** | **1.26** |

Table 1: Comparison between our model and baselines on all synthetic datasets and Hand-Me-That datasets for prediction tasks. Bold text represents the best result and underline denotes the second-best result.

For *real-world datasets*, we identified four interesting datasets that capture human behaviors, which are highly likely to be driven by human mental states. We have empirically or expert-knowledge-based defined realistic logic rule templates as prior knowledge. See Appendix. C and Appendix. D for dataset and prior knowledge details respectively. Followings are brief introduction to these real-world datasets: *i) Hand-Me-That* (Wan et al., 2022): contains multiple episodes of human-robot interactions in household tasks with a textual interface. We focus on the *change-state* type episodes and extract 503 sequences with average action trajectory length 30.5. *ii) Car-following* (Li et al., 2023): The data is extracted and enhanced from the open Lyft level-5 dataset (Houston et al., 2021) collected from urban and suburban environments along a fixed route in Palo Alto, California. We extract 2000 car-following behavior sequences with average action events 3.6 and average time horizon 19.44s. *iii) MultiTHUMOS* (Yeung et al., 2018): a dataset recording human actions extracting from videos. We focus only on the basketball dataset with 32 sequences. The time horizon of each sequence is 208.32s with 38.41 actions on average. *iv) EPIC-Kitchen-100* (Damen et al., 2018): a collection of first-people long-term unscripted activities in kitchen. We focus on two goals: cut onion and pour water, and extract 131 sequences contains related key actions. The time horizon of each sequence is 500s with 5.41 actions on average.

We want to emphasize that during the training process, only the action trajectories are given. For synthetic datasets, the ground truth mental events are known, allowing us to sample mental events and compare them with the ground truth. In the case of real-world datasets, the ground truth mental events are hidden, and thus we cannot directly compare the accuracy of sampled mental events. However, comparing the accuracy of action predictions is still feasible.

**Baselines** We choose several state-of-the-art baselines considering three different fields: *i) Neural Temporal Point Process Model*: RMTPP (Du et al., 2016), THP (Zuo et al., 2020), PromptTPP (Xue et al., 2023), and HYPRO (Xue et al., 2022), *ii) Logic-Based Model*: TELLER (Li et al., 2021) CLNN (Yan et al., 2023), STLR (Cao et al., 2024), *iii) Generative Model*: AVAE (Mehrasa et al.,

| Category | Model | Car-Follow | | MultiTHUMOS | | EPIC-Kitchen | |
|---|---|---|---|---|---|---|---|
| | | ER%↓ | MAE↓ | ER%↓ | MAE↓ | ER%↓ | MAE↓ |
| Neural TPP | RMTPP | 35.71% | 2.64 | 67.01% | 8.72 | 49.02% | 41.17 |
| | THP | 33.43% | 2.31 | 62.32% | 7.12 | 42.19% | 37.13 |
| | PromptTPP | 33.29% | 2.11 | 60.35% | 7.00 | 40.82% | 33.21 |
| | HYPRO | 32.86% | 2.03 | 58.25% | 6.98 | 42.28% | 35.98 |
| Rule-based Model | TELLER | 37.83% | 3.41 | 64.77% | 7.52 | 43.49% | 38.05 |
| | CLNN | 37.09% | 3.25 | 63.10% | 7.33 | 42.86% | 37.13 |
| | STLR | 32.75% | 2.47 | 63.38% | 7.69 | 43.37% | 36.85 |
| Gen. Model | AVAE | 35.08% | 2.95 | 61.17% | 8.32 | 43.56% | 39.24 |
| | GNTPP | 39.22% | 3.89 | 63.75% | 8.37 | 46.25% | 38.11 |
| | VEPP | 40.25% | 3.78 | 64.23% | 8.42 | 47.56% | 38.93 |
| | STVAE | 37.23% | 3.18 | 64.28% | 8.24 | 45.83% | 37.48 |
| – | **Ours\*** | **32.72%** | **1.80** | **57.20%** | **6.76** | **40.26%** | **32.19** |

Table 2: Comparison between our model and baselines on Car-Follow, MultiTHUMOS, and EPIC-Kitchen datasets for prediction tasks. Bold text represents the best result and underline denotes the second-best result.

2019), GNTPP (Lin et al., 2022), VEPP (Pan et al., 2020), and STVAE (Wang et al., 2023). For PromptTPP and HYPRO, in accordance with their setting, we choose AttNHP (Yang et al., 2021) as their base model, which is an attention-based auto-regressive model. For GNTPP, we choose the revised attentive history encoder and the temporal conditional VAE probabilistic decoder (Kingma & Welling, 2013). Detailed introduction for the baselines can be found in Appendix. E.

**Comparison Metric** Following common next-event prediction task in TPPs (Du et al., 2016; Zuo et al., 2020), our model as well as other baselines attempt to predict next event from history. Moreover, auto-regresively predicting multiple future events is also considered in our experiments. We evaluate the event type prediction with the Error Rate (ER%) and evaluate the event time prediction with the Mean Absolute Error (MAE).

## 5.2 EXPERIMENTS ON SYNTHETIC DATASET

**Infer Latent Mental Events and Learn Rule Parameters** Balancing inference accuracy and computational efficiency, we determined the resolution based on empirical results. Experiment results and corresponding analysis are detailed in Appendix. F. Results in Fig. 3 demonstrates a general alignment between the locations of fitted high occurrence probability and the actual occurrences of mental events. The mental events sampled based on these probabilities also correspond reasonably well to the actual time points of occurrence. The rule generator can correctly uncovers all ground truth rules given prior knowledge templates (as shown in Appendix. D) and decoder accurately learns the rule parameters. It

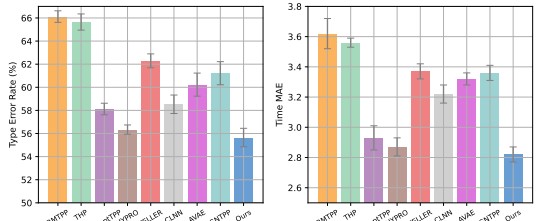

Figure 4: Performance of all the methods on predicting future 3 actions for Syn Data-2. **Left**: Comparison of event type average error rate ER%. **Right**: Comparison of event time average MAE.

is worth noting that our model effectively addresses the challenge of limited data by incorporating logical rules as prior knowledge. Even with a dataset size of only 2000 samples, it achieves promising results. Additionally, as we infer latent mental events through probability-based sampling, there is a certain degree of error involved. However, this error remains within an acceptable range.

**Next Single Event Prediction** We conduct the experiments on two synthetic datasets to predict the next single future action events. The experimental results are presented in Tab. 1, from which one can observe that our model outperforms all the baselines.

**Next Multiple Events Prediction** We also attempt to predict multiple actions from history. Auto-regressive long-horizon prediction may cause *cascading error* in TPP (Xue et al., 2022). The HYPRO method considered in the baselines is purely data-driven, which is a flexible neural-based model combined with expressive energy-based models. In contrast, our method employs a neu-

ral black-box encoder, but with a rule-based white-box decoder. The prediction results rely on the learned logic rules. In a nutshell, this design choice inherently trades off interpretability for model expressiveness. Shown in Fig. 4, in task of predicting next 3 actions, our model surprisingly achieves comparable and even lower ER% and MAE due to incorporation of latent mental events, logic rules, and backtracking mechanism.

## 5.3 EXPERIMENTS ON REAL-WORLD DATASET

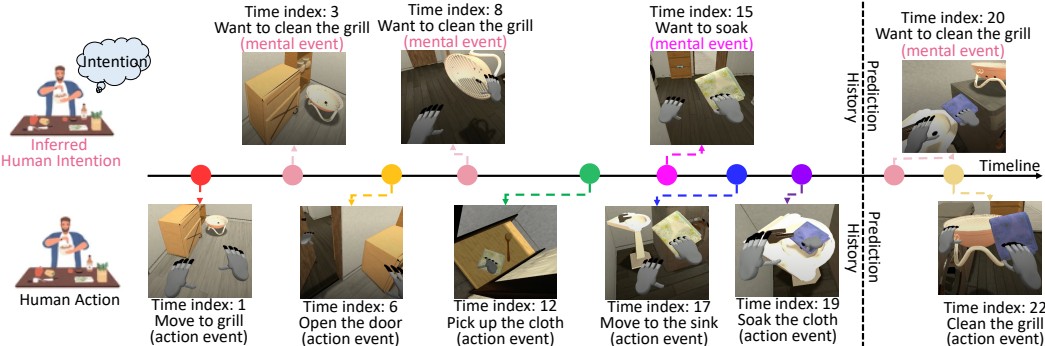

Figure 5: Inferred history mental events and predicted future events for one human trajectory aiming to clean the grill. **Top**: inferred and predicted human mental events. **Bottom**: observed and predicted human actions. Notice that the original dataset only includes the order of action occurrences. During data processing, we assume actions to have equal time intervals of 1 and then discretize the timeline with a resolution of 1. Then, we can use time indices to represent specific time point.

**Experiment Results**   On real-world datasets, we have designed prediction tasks that are specifically tailored to the characteristics of each dataset, taking into account the variations in the number of future events to be predicted. For the *Hand-Me-That*, *Car-Following*, and *EPIC-Kitchen-100* datasets, we focus on predicting the next action, whereas for the *MultiTHUMOS* dataset, we aim to predict the next 3 actions. The experimental results, presented in Tab. 1 and Tab. 2, demonstrate that our model performs exceptionally well in predicting both future event types and timings, outperforming all other methods.

**Prediction Examples**   Our model demonstrates intriguing applicability in real-life scenarios due to its ability of accurately predicting real-world events and speculating on human thoughts. As exemplified by the *Hand-Me-That* dataset, in Fig. 5, our proposed model effectively infers human historical intentions like *want to clean the grill*, and *want to soak*. It also correctly forecasts future human intentions and actions. In this instance, if the AI-Agent infers a person's intention to clean the grill at time index 20 and predicts that the person will clean the grill at time index 22, it can promptly retrieves the grill for him, which will significantly enhance convenience. Additionally, in the Appendix. G, we provide another prediction example using the *Car-Following* dataset.

## 5.4 SCALABILITY AND ABLATION STUDY

The scalability and computational efficiency is discussed in Appendix.H. The results show that our model can be effectively applied to large-scale datasets. Due to our restrictions on rule length and backtracking rounds, utilizing our model on large datasets will not excessively strain computational resources. We also conducted an ablation study to assess the importance of prior knowledge and backtracking module. The results are shown in Appendix. I, which confirm that appropriate prior knowledge enhances the accuracy of model predictions, and the inclusion of backtracking mechanism indeed improves model performance, even when there exists noise in the prior knowledge.

## 6 CONCLUSION

We propose a novel model for asynchronous action sequence modeling and inferring latent mental events utilizing a flexible encoder-decoder architecture that incorporates predefined temporal logic rule templates as prior knowledge and introduces a rule generator to refine them. The introduction of the backtracking mechanism improved the stability of the model's predictions. Our method demonstrates promising results in both synthetic and real-world datasets.

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

APPENDIX OVERVIEW

In the following, we will provide supplementary materials to better illustrate our methods and experiments.

- Section A provides the pseudocodes of the backtracking mechanism.
- Section B provides the details of rule generator.
- Section C provides comprehensive explanation of the generating process of synthetic datasets, along with an overview of the real-world datasets and the corresponding preprocessing details.
- Section D delineates the fundamental temporal logic rule templates as prior knowledge encompassing both the synthetic and real-world datasets. The rules governing the synthetic data are manually defined, whereas the rule templates for the real-world dataset are established through the utilization of experience and common knowledge.
- Section E introduces the baseline methods we considered in our paper.
- Section F analyses the effect of the discrete grid length on the fitted mental event occurrence probability.
- Section G provides more prediction examples on real-world dataset.
- Section H tests the scalability of our method using synthetic and real-world dataset with varying sample size.
- Section I conducts an ablation study to assess the importance of prior knowledge module ad the backtracking module.
- Section J provides the information of computing infrastructure for both synthetic data experiments and real-world data experiments.
- Section K provides the limitation and broader impacts of our paper.
- Section L provides an example of visualizing attention patterns for observed action sequence.

## A  BACKTRACKING MECHANISM

As shown in Alg. 1, we propose a backtracking mechanism when generating the next action to consistently reconstruct actions with relatively accurate samples of latent mental events. After the model is well-trained, this backtracking mechanism also plays a significant role in predicting future events.

## B  RULE GENERATOR

Our rule generating problem is solved via two alternating procedures: the master-problem and the sub-problem (Li et al., 2021), where the master-problem aims to re-weighting current rules, and the sub-problem is to search and construct a new temporal logic rule based on the given template. As describe in Sec. 3, both action and mental event can be the head predicate of temporal logic rules. We use $f_u$ to indicates the logic rule with $u$ being the head predicate and $\mathcal{F}$ to be the whole rule set. Denote $\mu_u$ as the head predicate specific base and $w_{f_u}$ is the rule specific weight. The joint likelihood given observed actions and inferred latent mental events can be calculated by Eq. (3), then we take log and denote the log-likelihood as $\ell(\boldsymbol{\mu}, \boldsymbol{w})$. We aim to uncover the set of temporal logic rules $\mathcal{F}$ via optimizing

$$Master\ Problem : \boldsymbol{\mu}^*, \boldsymbol{w}^* = \underset{\boldsymbol{\mu}, \boldsymbol{w}}{\arg\min} -\ell(\boldsymbol{\mu}, \boldsymbol{w}) + \Omega(\boldsymbol{w}), \quad s.t., \quad w_{f_u} \geq 0, \quad f_u \in \mathcal{F} \quad (16)$$

where $\Omega(\boldsymbol{w})$ is a convex regularization function that has a high value for "complexule" sets.

The above original problem is hard to solve, due to that the set of variables is exponentially large and can not be optimized simultaneously in a tractable way. We therefore start with a restricted master

---

**Algorithm 1** Backtracking Mechanism

---

**Input:** history actions $\boldsymbol{a}_{1:i-1}$, sampled mental events $\tilde{\boldsymbol{m}}_{1:\lfloor a_{i-1} \rfloor}$, pre-specified time horizon $T$, maximum number of backtracking $N$

**Output:** next action $\hat{a}_i$, sampled mental events between these two actions $\tilde{\boldsymbol{m}}_{\lfloor a_{i-1} \rfloor : \lfloor \hat{a}_i \rfloor}$

1: $\hat{t_i^a} = \int_{t_{i-1}^a}^{\infty} t \cdot p_{\boldsymbol{a}}(t | \boldsymbol{a}_{1:i-1}, \tilde{\boldsymbol{m}}_{\lfloor a_{i-1} \rfloor}) dt$

2: $\hat{k_i^a} = \arg\max \frac{\lambda_{k^a}(t | \boldsymbol{a}_{1:i-1}, \tilde{\boldsymbol{m}}_{\lfloor a_{i-1} \rfloor})}{\lambda_{\boldsymbol{a}}(t | \boldsymbol{a}_{1:i-1}, \tilde{\boldsymbol{m}}_{\lfloor a_{i-1} \rfloor})}$

3: $\hat{a}_i = (\hat{t_i^a}, \hat{k_i^a})$

4: **while** $\hat{t_i^a} < T$ **do**

5:     $\tilde{\boldsymbol{m}}_{1:\lfloor \hat{a}_i \rfloor} \sim \boldsymbol{q}_{\boldsymbol{\phi}, \lfloor \hat{a}_i \rfloor}(\boldsymbol{m})$

6:     **if** $\tilde{\boldsymbol{m}}_{\lfloor a_{i-1} \rfloor : \lfloor \hat{a}_i \rfloor} = \{\}$ **then**

7:         **return** $\hat{a}_i, \tilde{\boldsymbol{m}}_{\lfloor a_{i-1} \rfloor : \lfloor \hat{a}_i \rfloor}$                         # Case-1

8:     **else**

9:         $n = 0$

10:         **while** $n < N$ **do**

11:             Denote the first sampled mental event in $\tilde{\boldsymbol{m}}_{\lfloor a_{i-1} \rfloor : \lfloor \hat{a}_i \rfloor}$ as $\tilde{m}' = (\tilde{t}', \tilde{k}')$

12:             $\hat{t_i^a} = \int_{t_{i-1}^a}^{\infty} t \cdot p_{\boldsymbol{a}}(t | \boldsymbol{a}_{1:i-1}, \tilde{\boldsymbol{m}}_{\lfloor a_{i-1} \rfloor}, \tilde{m}') dt$

13:             $\hat{k_i^a} = \arg\max \frac{\lambda_{k^a}(t | \boldsymbol{a}_{1:i-1}, \tilde{\boldsymbol{m}}_{\lfloor a_{i-1} \rfloor}, \tilde{m}')}{\lambda_{\boldsymbol{a}}(t | \boldsymbol{a}_{1:i-1}, \tilde{\boldsymbol{m}}_{\lfloor a_{i-1} \rfloor}, \tilde{m}')}$

14:             $\hat{a}_i = (\hat{t_i^a}, \hat{k_j^a})$

15:             **if** $\hat{t_i^a} \leq \tilde{t}'$ and $\tilde{\boldsymbol{m}}_{\lfloor a_{i-1} \rfloor : \lfloor \hat{a}_i \rfloor} = \{\}$ **then**

16:                 **return** $\hat{a}_i, \tilde{\boldsymbol{m}}_{\lfloor a_{i-1} \rfloor : \lfloor \hat{a}_i \rfloor}$              # Case-2-1

17:             **else if** $\hat{t_i^a} > \tilde{t}'$ and $\tilde{\boldsymbol{m}}_{\lfloor \tilde{t}' \rfloor : \lfloor \hat{a}_i \rfloor} = \{\}$ **then**

18:                 **return** $\hat{a}_i, \tilde{m}', \tilde{\boldsymbol{m}}_{\lfloor \tilde{t}' \rfloor : \lfloor \hat{a}_i \rfloor}$          # Case-2-1

19:             **else**

20:                 Adopt $\tilde{m}'$ to history

21:                 **continue**

22:             **end if**

23:             $n = n + 1$

24:         **end while**

25:     **end if**

26: **end while**

27: **return** $\hat{a}_i, \tilde{\boldsymbol{m}}_{\lfloor a_{i-1} \rfloor : \lfloor \hat{a}_i \rfloor}$

---

problem (RMP), where the search space is much smaller, For example, we can start with an empty rule set, denoted as $\mathcal{F}_0 \subset \mathcal{F}$. Then we gradually expand this subset to improve the results, this will produce a nested sequence of subsets $\mathcal{F}_0 \subset \mathcal{F}_1 \subset ... \subset \mathcal{F}_k \subset ...$. For each $\mathcal{F}_k, k = 0, 1, ...,$ the restricted master problem is formulated by replacing the complete rule set $\mathcal{F}$ with $\mathcal{F}_k$:

$$RMP : \boldsymbol{\mu}^*_{(k)}, \boldsymbol{w}^*_{(k)} = \arg\min_{\boldsymbol{\mu}, \boldsymbol{w}} -\ell(\boldsymbol{\mu}, \boldsymbol{w}) + \Omega(\boldsymbol{w}), \quad s.t., \quad w_{f_u} \geq 0, \quad f_u \in \mathcal{F}_k \quad (17)$$

Solving the RMP corresponds to the evaluation of the current candidate rules. All rules in the current set will be reweighed (previously important rules may also be weighted down). The optimality of the current solution can be verified under the principle of the complementary slackness for convex problems, which in fact leads to the objective function of our subproblem. The optimal solution of the current RMP wile used to formulate the suboroblem. which is optimized to search for a new rule that may best improve the current likelihood.

A subproblem is formulated to propose a new temporal logic rule, which can potentially improve the optimal value of the RMP most. Given the current solution $\boldsymbol{\mu}^*_{(k)}, \boldsymbol{w}^*_{(k)}$, for the above RMP, a subproblem is formulated to minimize the increased gain.

$$Subproblem : \min_{\phi_{f_u}} -\frac{\partial \ell(\boldsymbol{\mu}, \boldsymbol{w})}{\partial w_{f_u}} + \frac{\Omega(\boldsymbol{w})}{\partial w_{f_u}} \Big|_{\boldsymbol{\mu}^*_{(k)}, \boldsymbol{w}^*_{(k)}} \quad (18)$$

Given prior knowledge temporal logic rule template, we consider add, remove predicates and corresponding temporal relations to refine the prior knowledge when solving sub-problem, which means that we do not need to evaluate rules constructed from scratch.

The rule generator module act as a plug-in module, with reliable prior knowledge and solid logical rules derived from it, we can freeze the rule generator module and solely rely on the prior knowledge to guide the generating process. During training of our encoder-decoder model, inferred latent mental events are paired with observed action events for rule mining. The generator is trained on this data until stable logic rules emerge. Hyper-parameters can be adjusted to balance accuracy and efficiency in rule learning. These rules then guide the decoder's generating process. Upon completion, rule bases and weights update synchronously with other model parameters.

## C  DATASETS

- **Synthetic Dataset**

  - *Syn Data-1*: 3 ground truth rules, 1 mental predicates and 2 action predicates. Each sequence has 18.60 actions on average.
  - *Syn Data-2*: a more complicated scenario with 4 ground truth rules, 2 mental predicates and 2 action predicates. Each sequence has 13.25 actions on average.

- **Real-World Dataset**

  - *Hand-Me-That* (Wan et al., 2022): contains 10,000 episodes of human-robot interactions in household tasks with a textual interface. In each episode, the robot first observes a trajectory of human actions towards her internal goal. Next, the robot receives a human instruction and takes actions to accomplish the subgoal behind the instruction. Here we consider robot's actions as the expert human trajectory. We combine human's history trajectory and robot's subsequent actions as a whole sequence from a single agent (human) and transfer the intermediate human instruction into human's mental state (e.g., human instruction: *Please soak the piece of cloth on the toilet* can be regard as human's mental: *want to soak the cloth*). The 10,000 episodes can be classified based on 3 human's instruction types: *bring-me*, *move-to* and *change-state*. Considering the diversity and practicability of defined logic rule templates, we focus on involving more action and mental predicates instead of complex objects' names, we mainly use change-state episodes. Abandoning episodes without human history trajectories, we finally get 503 sequences with average length 30.5.
  - *Car-Following* (Li et al., 2023): is processed from Lyft level-5 open dataset. The Lyft level-5 dataset(Houston et al., 2021) is a large-scale dataset of high-resolution sensor data collected by a fleet of 20 self-driving cars. The dataset includes 1000+ hours of perception and motion data collected over a 4-month period from urban and suburban environments along a fixed route in Palo Alto, California. The dataset covers diverse Car-Following(CF) regimes and the enhanced dataset provides smooth, ready-to-use motion information for Car-Following behaviors investigation. A regime refers to a driving situation experienced by the following vehicle (usually restricted by the leading vehicle). 29k+ Human Vehicle(HV)-following-Autonomous Vehicle(AV) pairs and 42k+ Human Vehicle(HV)-following-Human Vehicle(HV) pairs were selected and enhanced in similar environments from the Lyft level-5 dataset, with the total duration spanning over 460+ hours, covering a total distance of 15,000+ km. We mainly focus on HV-following-HV CF pairs because the essential information of HV-following-AV CF pairs(AV's speed and acceleration which are used to segment vehicle's regimes) were estimated by Kalman filtering, while all the information in HV-following-HV CF pairs are truly recorded with slight imputation of missing data. We extracted 2000 sequences from HV-following-HV CF pairs and defined 3 reasonable logic rule templates to explain the change of regimes in those sequences.
  - *MultiTHUMOS* (Yeung et al., 2018): a challenging dataset for action recognition, containing 400 videos of 65 different human actions. In this paper, we focus only on the basketball dataset with 32 sequences. The time horizon of each sequence is 208.32s with 38.41 actions on average.

- *EPIC-Kitchen-100* (Damen et al., 2018): a large-scale dataset in first-person (ego-centric) vision, which are multi-faceted, audio-visual, non-scripted recordings in native environments - i.e. the wearers' homes, capturing all daily activities in the kitchen over multiple days. In this paper, we focus on two goals in the kitchen: cut onion and pour water, and extract 131 sequences contains related key actions. The time horizon of each sequence is 500s with 5.41 actions on average.

## D    PRIOR KNOWLEDGE

For synthetic datasets, we manually designs temporal logic rule templates as the prior knowledge. And we also know the ground truth temporal logic rules for the synthetic datasets. For real-world datasets, we have defined a set of temporal logic rule templates as prior knowledge that align with intuition and experiential knowledge, capturing the time-based patterns associated with human mental intentions. It is noteworthy that these human mental related predicates are latent and do not actually exist within these datasets.

| Predicates | Explanation |
|:---:|:---:|
| $m_1$ | mental event-1 |
| $a_1$ | action event-1 |
| $a_2$ | action event-2 |

Table 3: Defined predicates and corresponding explanation for Syn data-1.

| Rule Num | Rule Content | Rule Weight |
|:---:|:---:|:---:|
| Rule-1 | $m_1 \leftarrow a_1, (a_1 \text{ before } m_1)$ | 0.6 |
| Rule-2 | $a_1 \leftarrow a_2, (a_2 \text{ before } a_1)$ | 0.6 |
| Rule-3 | $a_2 \leftarrow m_1, (m_1 \text{ before } a_2)$ | 0.8 |

Table 4: Ground truth temporal logic rules and corresponding weights for Syn Data-1.

| Rule Num | Rule Content |
|:---:|:---:|
| Rule-1 | $m_1 \leftarrow a_1, (a_1 \text{ none } m_1)$ |
| Rule-2 | $a_1 \leftarrow a_2, (a_2 \text{ none } a_1)$ |
| Rule-3 | $a_2 \leftarrow m_1, (m_1 \text{ none } a_2)$ |

Table 5: Temporal logic rule templates as prior knowledge for Syn Data-1.

- **Synthetic Dataset**
    - *Syn Data-1*: Defined predicates and ground truth temporal logic rules are shown in Tab. 3 and Tab. 4 respectively. The temporal logic rule templates provided as prior knowledge to the rule generator module of our model is shown in Tab. 5. Note that in the prior knowledge rule templates, the temporal relations are only "None". The prior knowledge temporal logic rule templates are partial and not entirely accurate, but capture some patterns of the ground truth rules. Our model aims to based on these kinds of prior knowledge to refine and generate more accurate logic rules. And we can compare the rule learning accuracy based on synthetic dataset, since we know the ground truth.
    - *Syn Data-2*: Defined predicates and ground truth temporal logic rules are shown in Tab. 6 and Tab. 7 respectively. The temporal logic rule templates provided as prior knowledge to the rule generator module of our model is shown in Tab. 8

- **Real-World Dataset**
    - *Hand-Me-That*: Extracted predicates and prior knowledge temporal logic rule templates are shown in Tab. 9 and Tab. 10 respectively.
    - *Car-Following*: Extracted predicates and prior knowledge temporal logic rule templates are shown in Tab. 11 and Tab. 12 respectively.

| Predicates | Explanation |
|:---:|:---:|
| $m_1$ | mental event-1 |
| $m_2$ | mental event-2 |
| $a_1$ | action event-1 |
| $a_2$ | action event-2 |

Table 6: Defined predicates and corresponding explanation for Syn data-2.

| Rule Num | Rule Content | Rule Weight |
|:---:|:---:|:---:|
| Rule-1 | $m_1 \leftarrow a_1, (a_1 \text{ before } m_1)$ | 0.6 |
| Rule-2 | $m_2 \leftarrow a_1 \wedge a_2, (a_1 \text{ before } a_2), (a_2 \text{ before } m_2)$ | 0.6 |
| Rule-3 | $a_1 \leftarrow m_2 \wedge a_2, (m_2 \text{ before } a_2), (a_2 \text{ before } a_1)$ | 1.0 |
| Rule-4 | $a_2 \leftarrow m_1, (m_1 \text{ before } a_2)$ | 1.0 |

Table 7: Ground truth temporal logic rules and corresponding weights for Syn Data-2.

- *MultiTHUMOS*: Extracted predicates and prior knowledge temporal logic rule templates in Tab. 13 and Tab. 14 respectively
- *EPIC-Kitchen-100*: Extracted predicates and prior knowledge temporal logic rule templates are defined in Tab. 15 and Tab. 16 respectively.

# E  BASELINES

In this paper, we primarily focus on baselines from three different fields: neural Temporal Point Process model, Logic-Based model, and generative model. Below, we will provide a detailed introduction to these baselines.

- **Neural Temporal Point Process Model**

    - RMTPP (Du et al., 2016): The approach considers the intensity function of a temporal point process as a nonlinear function that depends on the history. It utilizes a recurrent neural network to automatically learn a representation of the influences from the event history, which includes past events and time intervals, thereby fitting the intensity function of the temporal point process.

    - THP (Zuo et al., 2020): The model employs a concurrent self-attention module to embed historical events and generate hidden representations for discrete time stamps. These hidden representations are then used to model the interpolated continuous time intensity function. THP can also incorporate additional structural knowledge. Importantly, THP surpasses RNN-based approaches in terms of computational efficiency and the ability to capture long-term dependencies.

    - PromptTPP (Xue et al., 2023): The model incorporates a continuous-time retrieval prompt pool into the base TPP, enabling sequential learning of event streams without the need for buffering past examples or task-specific attributes. Specifically, this approach consists of a base TPP model, a pool of continuous-time retrieval prompts, and a prompt-event interaction layer. By addressing the challenges associated with modeling streaming event sequences, this mode enhances the model's performance.

    - HYPRO (Xue et al., 2022): The hybridly normalized probabilistic (HYPRO) model is capable of making long-horizon predictions for event sequences. This model consists of two modules: the first module is an auto-regressive base TPP model that generates prediction proposals, while the second module is an energy function that assigns weights to the proposals, prioritizing more realistic predictions with higher probabilities. This design effectively mitigates the cascading errors commonly experienced by auto-regressive TPP models in prediction tasks, thereby improving the model's accuracy in long-term forecasting.

- **Logic-Based Model**

| Rule Num | Rule Content |
|---|---|
| Rule-1 | $m_1 \leftarrow a_1, (a_1 \text{ none } m_1)$ |
| Rule-2 | $m_2 \leftarrow a_2, (a_2 \text{ before } m_2)$ |
| Rule-3 | $a_1 \leftarrow a_2, (a_2 \text{ before } a_1)$ |
| Rule-4 | $a_2 \leftarrow m_1, (m_1 \text{ none } a_2)$ |

Table 8: Temporal logic rule templates as prior knowledge for Syn Data-2.

| Predicates | Explanation |
|---|---|
| MoveTo | Move to a location or an object |
| PickUp | Pick up an object from a location or a receptacle |
| Put | Put an object on a location or into a receptacle |
| ToggleOn | Toggle on toggleable-thing, like electric device |
| Soak | Soak an object |
| Open | Open openable thing, like cabinet |
| Clean | Clean an object or a location |
| Cool | Freeze food |
| Slice | Slice food |
| Heat | Heat food |
| Close | Close openable thing |
| WantToPickUp | Want to get an object |
| WantToSoak | Want to soak an object |
| WantToOpenToGet | Want to open an openable thing to get an object |
| WantToToggleOn | Want to toggle on an electric device |
| WantToPut | Want to put an object on a location or into a receptacle |
| WantToClean | Want to clean a location or an object |
| WantToHeat | Want to heat food |
| WantToCool | Want to freeze food |
| WantToSlice | Want to slice an object |

Table 9: Defined predicates and corresponding explanation for Hand-Me-That dataset.

– TELLER (Li et al., 2021): It is a non-differentiable algorithm that can be described as a temporal logic rule learning algorithm based on column generation principles. This method formulates the process of discovering rules from noisy event data as a maximum likelihood problem. It also designs a tractable branch-and-price algorithm to systematically search for new rules and expand existing ones. The algorithm alternates between a rule generation stage and a rule evaluation stage, gradually uncovering the most significant set of logic rules within a predefined time limit.

– CLNN (Yan et al., 2023): The model learns weighted clock logic (wCL) formulas, which serve as interpretable temporal logic rules indicating how certain events can promote or inhibit others. Specifically, the CLNN model captures temporal relations between events through conditional intensity rates guided by a set of wCL formulas that offer greater expressiveness. In contrast to conventional approaches that rely on computationally expensive combinatorial optimization to search for generative rules, CLNN employs smooth activation functions for the components of wCL formulas. This enables a continuous relaxation of the discrete search space and facilitates efficient learning of wCL formulas using gradient-based methods.

– STLR (Cao et al., 2024): A model specifically designed for learning spatial-temporal logic rules in order to explain human actions. It consists of two main modules: the rule generator, employing the transformer to infer logic rules by treating them as latent variables, and the reasoning evaluator, which predicts future entity trajectories based on the generated rules. While STLR demonstrates flexibility in generating logic rules without relying on prior knowledge, it lacks the ability to infer fine-grained latent mental events in a real-time manner, a capability inherent in our method.

• **Generative Model**

| Rule Num | Rule Content |
|---|---|
| Rule-1 | PickUp ← WantToPickUp ∧ MoveTo, (WantToPickUp before MoveTo) |
| Rule-2 | Soak ← WantToSoak ∧ MoveTo ∧ Put ∧ ToggleOn, (WantToSoak before MoveTo), (MoveTo before Put), (Put before ToggleOn) |
| Rule-3 | PickUp ← WantToOpenToGet ∧ MoveTo ∧ Open ∧ (WantToOpenToGet before MoveTo), (MoveTo before Open) |
| Rule-4 | ToggleOn ← WantToToggleOn ∧ MoveTo, (WantToToggleOn before MoveTo) |
| Rule-5 | Put ← WantToPut ∧ MoveTo, (WantToPut before MoveTo) |
| Rule-6 | Clean ← WantToClean ∧ Soak ∧ PickUp ∧ MoveTo, (WantToClean before Soak), (Soak before PickUp), (PickUp before MoveTo) |
| Rule-7 | Heat ← WantToHeat ∧ PickUp ∧ MoveTo ∧ Put ∧ ToggleOn, (WantToHeat before PickUp), (PickUp before MoveTo), (MoveTo before Put), (Put before ToggleOn) |
| Rule-8 | Cool ← WantToCool ∧ PickUp ∧ MoveTo ∧ Open ∧ Put ∧ Close, (WantToCool before PickUp), (PickUp before MoveTo), (MoveTo before Open), (Open before Put), (Put before Close) |
| Rule-9 | Slice ← WantToSlice ∧ Put ∧ PickUp, (WantToSlice before Put), (Put before PickUp) |

Table 10: Temporal logic rules as prior knowledge for Hand-Me-That dataset.

| Predicates | Explanation |
|---|---|
| Fa | Free acceleration |
| C | Cruising at a desired speed |
| A | Acceleration following a leading vehicle |
| D | Deceleration following a leading vehicle |
| F | Constant speed following |
| ConservativeIntention | The driver has a conservative intention, maintaining their speed |
| AggressiveIntention | The driver has an aggressive intention, tending to accelerate |

Table 11: Defined following cars' predicates and corresponding explanation in Car-Following dataset.

- AVAE (Mehrasa et al., 2019): The model is a recurrent variational auto-encoder designed for modeling asynchronous action sequences. At each time step, the model utilizes the history of actions and inter-arrival times to generate a distribution over latent variables. A sample from this distribution is then decoded into probability distributions for the inter-arrival time and action label of the next action. To address the limitations of using a fixed prior in the traditional VAE framework, this model incorporates a prior net that enhances the learning process.

- GNTPP (Lin et al., 2022): The model is a comprehensive generative framework for neural temporal point process modeling. It utilizes deep generative models as probabilistic decoders to approximate the target distribution of occurrence time. For the encoder, the model considers both RNN-based methods and self-attention-based mechanisms. As for the decoder, the model incorporates multiple generative models, such as the temporal conditional diffusion denoising model, temporal conditional VAE model, temporal conditional GAN model, temporal conditional continuous normalizing flow model, and temporal conditional noise score network model. The various combinations of encoders and decoders make the GNTPP highly flexible.

- VEPP (Pan et al., 2020): The model employs LSTM to embed the event sequence and utilizes VAE for modeling the event sequence. It leverages the latent information to capture the distribution over the event sequence.

- STVAE (Wang et al., 2023): A probabilistic model based on the variational temporal point process to synthesize human trajectories.

| Rule Num | Rule Content |
|---|---|
| Rule-1 | A ← C ∧ AggressiveIntention, (C before AggressiveIntention) |
| Rule-2 | C ← Fa ∧ ConservativeIntention, (Fa before ConservativeIntention) |
| Rule-3 | F ← A ∧ D ∧ ConservativeIntention, (A before D), (D before ConservativeIntention) |

Table 12: Temporal logic rules as prior knowledge for Car-Following dataset.

| Predicates | Explanation |
|---|---|
| Dribble | Dribbling the basketball |
| Pass | Passing the basketball from one person to another |
| Shot | An attempt to put the basketball in the basketball hoop |
| PoorShootingOpportunity | Mental event, a player think that this is not a good shooting opportunity |
| GoodShootingOpportunity | Mental event, a player think that this is a good shooting opportunity |

Table 13: Defined predicates and corresponding explanation for MultiTHUMOS basketball dataset.

# F ANALYSIS OF GRID LENGTH

In order to investigate the impact of the selected time interval size (time grid length) for discretizing the timeline on fitting the probability of mental event occurrences, we conducted the following experiment on Syn Data-1 with known mental event scenarios. As described in the paper, we still employ the self-attention module to embed all historical action events, and utilize LSTM to fit the probability of mental event occurrences on each time grid. However, in this experiment, we only consider the likelihood of mental event occurrences and assume that the mental events are given during the calculation process. Since in this scenario we know the actual occurrence of the mental event on a specific time grid, the likelihood of the mental event is determined by the product of the probabilities of the mental event occurring on its true grid and the complement of the probabilities of the mental event not occurring on these grids. Therefore, the likelihood for mental process is given by,

$$\mathcal{L}_{m_i} = \prod_{\xi=1}^{L} \left( \mathbb{P}\left((t_j^m, k_j^m) \in V_\xi\right) \cdot \mathbb{I}(m_i \text{ actually occurs in } V_\xi) \right)$$
$$\cdot \left( \left(1 - \mathbb{P}\left((t_j^m, k_j^m) \in V_\xi\right)\right) \cdot \mathbb{I}(m_i \text{ actually not occurs in } V_\xi) \right) \quad (19)$$

where $(t_j^m, k_j^m)$ is the occurrence of mental event $m_i$ and $k \in \{1, ..., |\mathcal{M}|\}$, $V_\xi$ is the $\xi$-th time grid. $\mathbb{I}(\cdot)$ is the indicator function that takes a value of 1 when the condition is satisfied, and 0 otherwise.

Accordingly, the log-likelihood function is given by,

$$\log \mathcal{L}_{m_i} = \sum_{\xi=1}^{L} \log\left(\left(\mathbb{P}\left((t_j^m, k_j^m) \in V_\xi\right) \cdot \mathbb{I}(m_i \text{ actually occurs in } V_\xi)\right)\right.$$
$$\left. \cdot \left(\left(1 - \mathbb{P}\left((t_j^m, k_j^m) \in V_\xi\right)\right) \cdot \mathbb{I}(m_i \text{ actually not occurs in } V_\xi)\right)\right) \quad (20)$$

Then we only train the encoder via optimizing the log-likelihood for mental process. We conducted tests on five different grid lengths, namely 0.05, 0.10, 0.20, 0.25, and 0.50, in the dataset with a time horizon of 15, which divide the entire time horizon into 300, 150, 75, 60, and 30 small grids respectively. For illustrative purposes, we have selected two sequences to showcase the results of fitting the probability of mental event occurrences. From Fig. 6, it is evident that after the convergence of log-likelihood, in experiments with different grid lengths, the areas with higher fitted probabilities closely align with the actual grids where the mental events occur. However, as the grid length decreases, the corresponding fitted probabilities also decrease.

Therefore, the grid length has a relatively minor impact on fitting the probability of mental event occurrences. However, larger grid lengths result in fewer sampling instances, which helps to reduce randomness when our model's encoder samples based on the fitted probabilities of mental event occurrences. Additionally, the reduced number of sampling instances also contributes to improving the training efficiency of our model.

| Rule num | Rule Content |
|----------|--------------|
| Rule-1 | Dribble ← PoorShootingOpportunity, (PoorShootingOpportunity before Dribble) |
| Rule-2 | Pass ← PoorShootingOpportunity, (PoorShootingOpportunity before Pass) |
| Rule-3 | Shot ← GoodShootingOpportunity, (GoodShootingOpportunity before Shot) |

Table 14: Temporal logic rules as prior knowledge for MultiTHUMOS basketball dataset.

| Predicates | Explanation |
|------------|-------------|
| TakePlate | Retrieve the plate for future use |
| TakeEggs | Retrieve the eggs for further use |
| TakeOnion | Retrieve the onion for further use |
| TakeGlass | Retrieve the glass for further use |
| CutOnion | Retrieve the onion for further use |
| PourWater | Pour water to glass |
| NeedOnionToCook | Mental event, one has the intention to use onion to cook |
| NeedWater | Mental event, one needs water in kitchen |

Table 15: Defined predicates and corresponding explanation for EPIC Kitchen dataset.

## G    MORE PREDICTION EXAMPLE ON REAL-WORLD DATASET

We provide another prediction example for *Car-Following* dataset. In Fig. 7, our model infers a driver's historical intentions and predicts their future car-following actions. In the field of autonomous driving, a self-driving vehicle can adjust its lane and drive reasonably by considering the inferred intentions of human drivers in neighboring lanes and their predicted behaviors, all while adhering to traffic regulations.

## H    SCALABILITY

To test the scalability of our proposed model, we have added two more synthetic datasets with more complex ground truth rules and larger domains for latent mental states (Syn Data-3: 5 ground truth rules, 4 latent mental states. Syn Data-4: 6 ground truth rules, 6 latent mental states). And we have added versions of the new datasets with sample sizes from 1000 to 5000 to study the scalability of our model. We have also extracted more data sequences for Hand-Me-That and Car-Following datasets to investigate the scalability of our model to handle large-scale real-world datasets. As depicted in Fig. 8, our model swiftly converges with acceptable runtime even with large-scale datasets. The model's prediction performance improves with larger sample sizes, demonstrating its good scalability.

For more intricate rules and mental space domains like Syn Data-4, the prediction ER% decreased to 52.08% and the MAE reduced to 2.52 when provided with 5000 samples, highlighting its capability to handle complex domains.

For these two real-world datasets with different sample sizes, the prediction performance improved with more samples and also resulted in more computational cost. For small-scale real-world dataset, our model adeptly handles challenges posed by small datasets. Even with a dataset size of only 1000 samples, our model delivers satisfactory results. For the Hand-Me-That dataset with 1000 samples, the ER% and MAE are 73.25% and 1.22. For the Car-Following dataset with 1000 samples, the ER% and MAE are 35.38% and 2.12, which are comparable with the majority baselines trained with larger sample size. For large-scale real-world dataset, the ER% and MAE decrease to 68.19% and 1.13 for Hand-Me-That dataset with 5000 samples. And these two metrics decrease to 30.18% and 1.69 for the Car-Following dataset with 5000 samples. Even on large-scale datasets, our algorithm converges relatively quickly with current computational infrastructure, indicating the ability of our proposed model to handle large-scale real-world datasets. This is attributed to our backtracking mechanism's ability to adjust the number of backtracking rounds as the data volume grows, and the constraints of rule length to reduce search space, thereby mitigating computational complexity to some extent.

| Rule num | Rule Content |
|---|---|
| Rule-1 | CutOnion ← NeedOnionToCook ∧ TakeOnion, 
 (NeedOnionToCook before TakeOnion) 
 (NeedOnionToCook before CutOnion) 
 (TakeOnion before CutOnion) |
| Rule-2 | PourWater ← NeedWater ∧ TakeGlass, 
 (NeedWater before TakeGlass) 
 (NeedWater before PourWater) 
 (TakeGlass before PourWater) |

Table 16: Temporal logic rules as prior knowledge for EPIC Kitchen dataset.

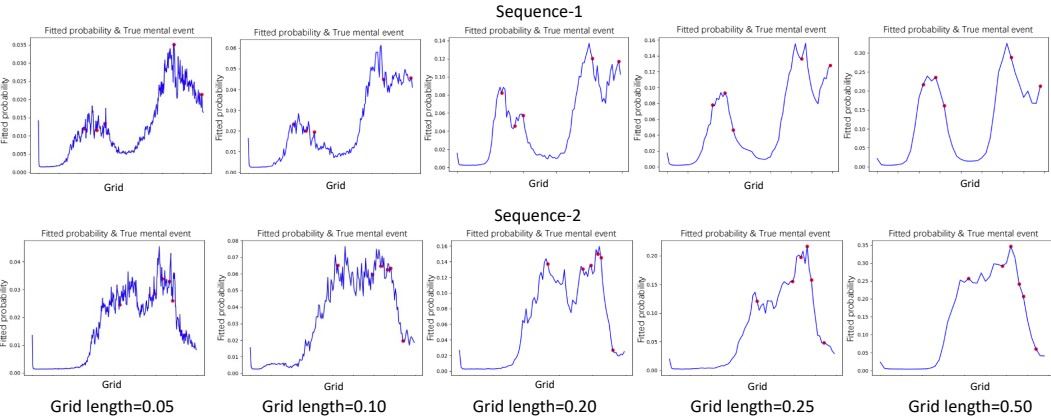

Figure 6: Experiments for examining the impact of different grid lengths on the fitted probability of mental event occurrences. The blue line represents the fitted probability of mental event occurrences, while the red points indicate the ground truth occurrences of mental events.

Across all these datasets, model's prediction performance enhances as sample sizes increase, while maintaining acceptable time costs, showcasing its good scalability.

## I ABLATION STUDY

We conducted an ablation study to assess the importance of different components, using the following ablation settings: (i) removing the prior knowledge and removing the backtracking module, (ii) solely removing the backtracking module, (iii) solely removing the prior knowledge module, (iv) evaluating the full model as described in our paper. To evaluate the impact of the prior knowledge module, it is necessary to have access to the ground truth logic rules. Consequently, our ablation study is performed on synthetic datasets. The results are shown in Tab. 17:

| Ablation Settings | | Syn Data-1 | | Syn Data-2 | |
|---|---|---|---|---|---|
| Prior Knowledge | Backtracking | ER(%) | MAE | ER(%) | MAE |
| No | No | 48.28% | 3.1275 | 52.50% | 3.2378 |
| Yes | No | 45.64% | 2.6438 | 48.37% | 2.8645 |
| No | Yes | 43.76% | 2.5763 | 47.93% | 2.7247 |
| Yes | Yes | **41.72%** | **2.3182** | **46.85%** | **2.5192** |

Table 17: Ablation study on synthetic datasets.

If we exclude the prior knowledge module, it is important to note that the rule generator will commence with an empty rule set. Consequently, the training process will require additional time to converge due to increased iterations between solving the master problem and solving the sub problems within the rule generator module.

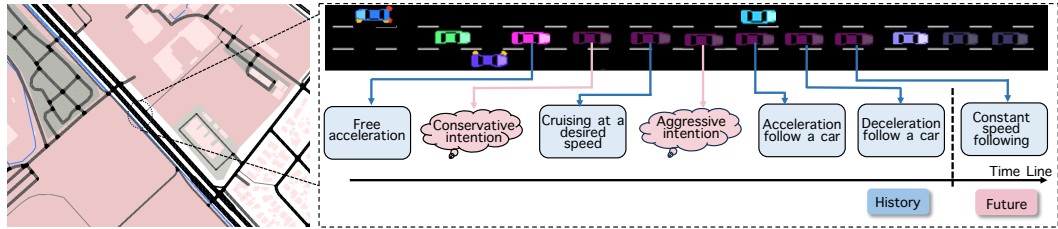

Figure 7: **Left:** Satellite map of Palo Alto, California, extracted from Google Earth (Goo, 2022), **Right:** Car following process (pink car) for one car trajectory. The historical mental event inferred by our method is indicated within the pink boxes. The next action in the future of the pink car predicted by our method is represented in the blue box to the right of the dashed line. The visualization is enhanced via SUMO simulator (Krajzewicz et al., 2002; Song et al., 2014).

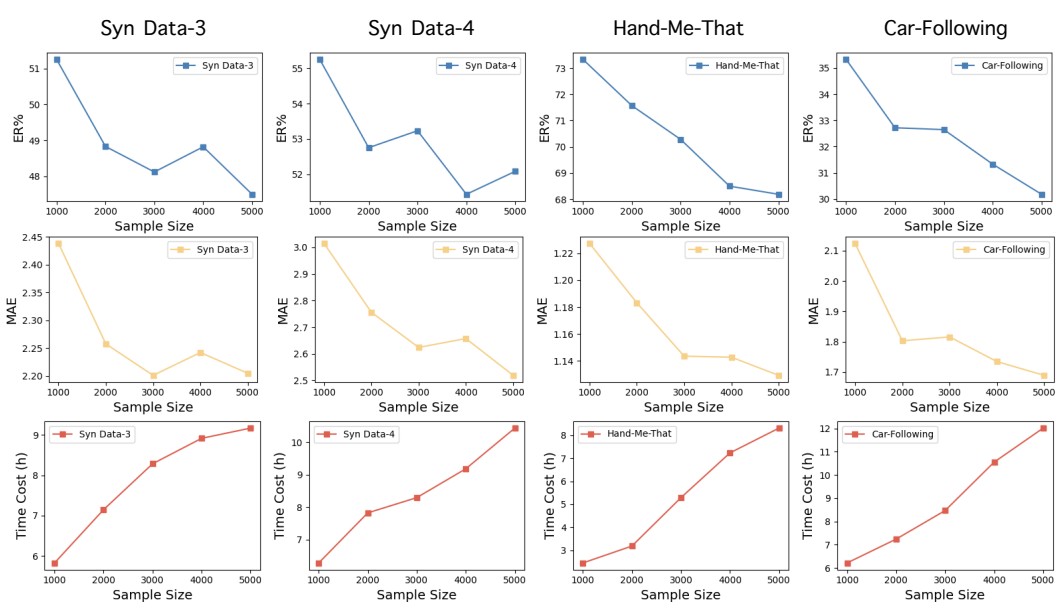

Figure 8: Scalability and the computation time cost of our method. Syn Data-3 (5 ground truth rules, 4 latent mental states) and Syn Data-4 (6 ground truth rules, 6 latent mental states) are newly added synthetic datasets. Hand-Me-That and Car-Following are the real-world dataset we used in paper but we extract more samples. For each dataset, we vary the sample size from small to large scale.

The experiments were conducted with consistent settings and hyperparameters as described in our paper. The results from different ablation settings confirm that appropriate prior knowledge enhances the accuracy of model predictions. Additionally, the inclusion of a backtracking mechanism plays a more vital role which significantly improves model performance, even when there is some level of noise in the prior knowledge. Overall, both modules contribute to enhancing the model's efficiency.

## J    COMPUTING INFRASTRUCTURE

All synthetic data experiments, as well as the real-world data experiments, including the comparison experiments with baselines, are performed on Ubuntu 20.04.3 LTS system with Intel(R) Xeon(R) Gold 6248R CPU @ 3.00GHz, 227 Gigabyte memory.

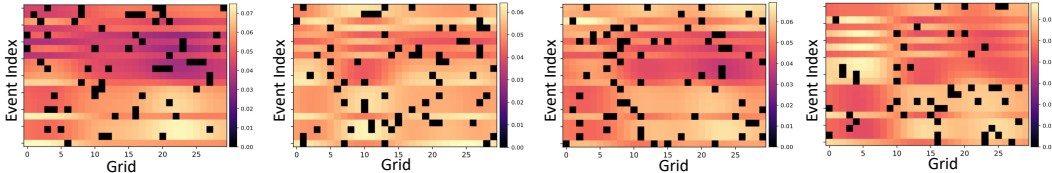

Figure 9: Visualization of attention patterns of different attention heads for action sequence in Syn Data-1 on different discrete time grids.

## K  LIMITATION AND BROADER IMPACTS

Our work has vast potential applications in the field of human-robot collaboration. Our approach enables timely and accurate inference of human mental events, as well as precise prediction of future human behavior. This will assist robots in providing timely, accurate, and useful assistance. For instance, it can aid elderly individuals with limited mobility in managing daily activities or help self-driving vehicles navigate roads more safely and smoothly.

One limitation of our current approach is the reliance on hand-crafted logic rule templates in the decoder. While these rules provide interpretability, they may introduce biases and limit the model's flexibility. To mitigate this limitation, we could explore techniques for automatically learning rules from data without prior knowledge logic rule templates. By leveraging advanced machine learning algorithms, such as reinforcement learning or differentiable logic programming, we can train the model to discover and refine rules directly from the observed data. This approach would enhance the model's flexibility by adapting to the nuances of the data and reduce the risk associated with manually introduced biases.

Additionally, the discretization of the timeline might introduce some noise when sampling the latent mental events. In real-world scenarios, establishing a well-defined and fine-grained discrete time grid can necessitate conducting numerous experiments. It's worth noting that choosing the interval for discretization is a tunable hyperparameter. We can explore methods to automate hyperparameter tuning to streamline this process and ensure optimal performance without the need for extensive manual experimentation.

## L  ILLUSTRATION OF ATTENTION WEIGHT FOR OBSERVED ACTIONS

In our proposed model, the temporal point processes involve triggering between latent mental and action events, where historical actions can influence latent events and vice versa. Therefore, we resort to the attention mechanism to map the information of the entire action sequence on the spanning time horizon on each discrete time grid. But the attention in our model cannot capture how the mental state influences actions. The influence of mental process on action process is reflected on the intensity function after the inference of latent mental process.

In Fig.9, we provided an example of attention weights for action sequence of Syn Data-1 on different discrete time grids, which visualizes attention patterns of different attention heads. Pixel $(i, \xi)$ in each figure signifies the attention weight of the event $(t_i^a, k_i^a)$ attending to the discrete time grid $t_\xi$. We can see that each attention head employs a different pattern to capture dependencies. For each attention head, the impact of one event is different on each discrete time grid, reflected by various attention weights. The impact of the entire action sequence on a discrete time grid can be conceptualized as a weighted combination of the entire action sequence, with weights derived through attention mechanisms. The attention mechanism effectively captures the potential influence of historical events at specific time points on the intensity of future events.

