# OpenReview forum: "Inference of Evolving Mental States from Irregular Action Events to Understand Human Behaviors"
_ICLR.cc/2025/Conference — Submitted to ICLR 2025_

### Official Review · Reviewer_1Fue · 2024-10-29

**Soundness:** 3
**Presentation:** 2
**Contribution:** 2
**Rating:** 6
**Confidence:** 4

**Summary:**

The paper introduces an versatile encoder-decoder framework to infer and predict evolving human mental states from irregular action events. By implementing a temporal point process model, the authors handle the irregular nature of both action and mental data. A key enhancement is integrating a backtracking mechanism in the decoder, improving the accuracy of future action and mental state predictions. The model also incorporates logic rules as priors, providing robustness in scenarios with limited data availability.

**Strengths:**

This paper proposes a novel model for asynchronous action sequence modeling and inferring latent mental events. It utilizes a flexible encoder-decoder architecture incorporating predefined temporal logic rule templates as prior knowledge and introducing a rule generator to refine them. The introduction of the backtracking mechanism improved the stability of the model’s predictions. This method demonstrates promising results in both synthetic and real-world datasets. While the integration of rule-based logic for dynamic prediction adjustment presents an interesting angle, this concept may echo principles already employed in fields like imitation learning, where actions are predicted based on observed sequences and thought cloning that attempts to replicate cognitive processes.

**Weaknesses:**

W1. Encoder-decoder architectures and the inference of mental states have been explored in adjacent areas such as imitation learning and thought cloning. These methodologies similarly attempt to capture and reproduce complex patterns of behavior or cognitive states based on observed actions or external stimuli. How does the proposed model's use of temporal point processes and rule-based logic distinctly improve upon or differ from similar mechanisms used in imitation learning or thought cloning? Can you provide specific examples or comparative analyses that highlight these differences?

W2. The paper does not discuss the computational demands and scalability of the proposed model, which is crucial for assessing its practical applicability in real-world scenarios.

W3. The model's dependence on manually defined logic rules limits its flexibility and may introduce potential biases, reducing its ability to adapt to new data.

W4. In the experimental section, the authors validated their method's effectiveness on both synthetic and real datasets. Although ER% and MAE provide basic quantitative metrics for model performance, considering more evaluation dimensions and a more detailed experimental design could offer deeper insights when dealing with complex psychological prediction models. Particularly, the method seems very similar to thought cloning, and more evaluation metrics should be considered for real datasets like Hand-Me-That to validate the method's practical effectiveness. Here are some references:

[1] Thought Cloning: Learning to Think while Acting by Imitating Human Thinking

[2] Enhaning Human-AI Collaboration Through Logic-guided reasoning

[3] Infer Human’s Intentions Before Following Natural Language Instructions

W5. Although the model of mental states and action events is introduced, it lacks a detailed explanation of how sub-goals or mental states affect the generation of action events in long horizon tasks, or how action events feedback influences the prediction of mental states. This missing or unclear interaction could make the model's theory perplexing.

**Questions:**

Q1. The author mentioned, "A novel backtracking action sampling mechanism iteratively refines predictions, significantly improving responsiveness to real-time fluctuations in human behavior." However, in the decoder (Section 4.2), the column generation algorithm involves frequent rule updates and optimizations, which may lead to high computational costs and execution times, especially when the rule set is large or the data volume is substantial. The backtracking mechanism (Section 4.4) sounds innovative, but repeatedly adjusting previously inferred mental states may also require significant computational resources, particularly in dynamic and changing environments. Therefore, could the authors include a comparison of computational costs in the experimental section to assess how well this mechanism performs in real-time applications?

Q2. In Formula 13, the authors use the Monte Carlo method to estimate expectations, which may introduce bias, especially when the sample size is insufficient. Is there any theoretical or empirical analysis to address this issue?

Q3. In Tables 4 and 7, the authors presented ground truth temporal logic rules and corresponding weights for Syn Data-1 and Syn Data-2. How are the rule weights determined? Are they predefined or calculated? Why are they 0.6 or 0.8, and how do these rule weights impact the results?

Q4. The preliminaries mention the use of DT-RP to model mental states in discrete time, while the generation of action events is simulated in continuous time by TPP. Could the simultaneous use of discrete and continuous time in the model possibly lead to inconsistencies in time scales? How do the methods ensure effective transition and interfacing between these two-time scales?

Q5. Although the model of mental states and action events is introduced, it lacks a detailed explanation of how sub-goals or mental states affect the generation of action events in long horizon tasks, or how action events feedback influences the prediction of mental states. This missing or unclear interaction could make the model's theory perplexing.

Q6. Although self-attention can handle action sequences, its decision-making process is usually not transparent, with poor interpretability. Could techniques like visualization of attention weights be employed to explain how the encoder's mental state influences actions, especially when actions are frequent, or are there anomalies in the sequence?

---

> ### Author Response · Authors · 2024-11-22
> **Response to Reviewer 1Fue**
>
> We first thank reviewer 1Fue for the meticulous analysis and insightful comments, especially for the questions about details of our model and potential extension, which benefit us to further clarify our paper. We would like to address the concerns one by one.
>
> In response to your mentioned weaknesses:
>
> **W1**: Compared to traditional imitation learning models, our proposed model integrates human mental processes into behavior generation. In contrast to thought clone models that rely on **supervised learning** and necessitate actual human thought data for training, our method enables the inference of unobserved mental processes from observed human actions in an **unsupervised manner**. Our approach eliminates the need for human thought data, which is often difficult to acquire in real-world scenarios.
>
> Moreover, both imitation learning and thought cloning typically operate in **discrete-time** settings, neglecting the irregular occurrence nature of human thought processes reflected in irregular event-to-event time intervals. Our model, on the other hand, addresses mental event inference in a **continuous-time** framework, capturing the evolving nature of these processes over time.
>
> In terms of interpretability, our approach, with the incorporation of logic rules, excels in explaining the intricate triggering mechanisms between mental processes and action processes. In contrast, imitation learning and thought cloning models lack this capability.
>
> **W2**: In the revised paper, we have outlined the computational requirements in Appendix J and presented scalability and computational cost experiments in Appendix H. In scalability experiments, we have expanded our evaluation to include two additional synthetic datasets and two real-world datasets with varying sample sizes. The new synthetic datasets encompass intricate rules and enlarged latent state domains. Additionally, we augmented the data sequences for the Hand-Me-That and Car-Following datasets to assess our model's scalability to handle extensive real-world datasets. Here, we present our experimental results from both small and large datasets to showcase our findings.
>
> For **small-scale real-world dataset**, our model adeptly handles challenges posed by small datasets. Even with a dataset size of only 1000 samples, our model delivers satisfactory results. For the Hand-Me-That dataset with 1000 samples, the ER% and MAE are 73.25% and 1.22. For the Car-Following dataset with 1000 samples, the ER% and MAE are 35.38% and 2.12, which are comparable with the majority baselines trained with larger sample size.
>
> For **large-scale real-world dataset**, the ER% and MAE decrease to 68.19% and 1.13 for Hand-Me-That dataset with 5000 samples. And these two metrics decrease to 30.18% and 1.69 for the Car-Following dataset with 5000 samples. Even on large-scale datasets, our algorithm converges relatively quickly with current computational infrastructure, indicating the ability of our proposed model to handle large-scale real-world datasets. This is attributed to our backtracking mechanism's ability to adjust the number of backtracking rounds as the data volume grows, and the constraints of rule length to reduce search space, thereby mitigating computational complexity to some extent.
>
> Across all four datasets, model's prediction performance enhances as sample sizes increase, while maintaining acceptable time costs, showcasing its good scalability.
>
> **W3**: Although we incorporate rule templates as priors to utilize rich domain knowledge of the application, our rule-generation algorithm is inherently capable of learning rules from scratch, even in scenarios with limited prior knowledge. By leveraging column-generation framework, the algorithm can autonomously discover and refine new rules, with its accuracy improving as more data becomes available. Additionally, standard techniques such as constraining rule lengths ensure computational efficiency and prevent an excessively large search space.
>
> In practice, we can employ two complementary strategies tailored to different problem settings: autonomous rule learning for data-rich domains and template-guided learning for scenarios with limited data but abundant prior knowledge.
>
> In our response to **Reviewer orep, W7**, we have evaluated the effectiveness of our model in scenarios with and without predefined rules, across different sample sizes. Please refer to the table for detailed experiment results and corresponding analysis. Briefly, with same training samples, not providing prior rule templates during training leads to basically satisfactory model performance, but increased training time. When increasing the training sample size without prior rule templates, the model exhibits a notable improvement in converged negative loss and prediction accuracy, approaching or even matching the model performance achieved with prior rule templates on a smaller dataset, showing good ability to adapt to data lacking prior knowledge.

---

> ### Author Response · Authors · 2024-11-22
> **Response to Reviewer 1Fue**
>
> **Continuing with W3**: Moreover, many works also extract prior logic rules from expert knowledge, regulations, or statistical findings to guide models across various domains and tackle challenges posed by limited data [1, 2, 3, 4]. In high-stakes scenarios, incorporating prior knowledge enables our model to align well with human cognition and learning processes and tackle limited data issues.
>
> [1] Liu, J., Wang, H., Cao, Z., Yu, W., Zhao, C., Zhao, D., ... & Li, J. (2023). Semantic traffic law adaptive decision-making for self-driving vehicles. IEEE Transactions on Intelligent Transportation Systems.\
> [2] Li, Shuang, et al. "Temporal logic point processes." International Conference on Machine Learning. PMLR, 2020.\
> [3] Zhang, Yizhou, Karishma Sharma, and Yan Liu. "Vigdet: Knowledge informed neural temporal point process for coordination detection on social media." Advances in Neural Information Processing Systems 34 (2021): 3218-3231.\
> [4] Marinescu, Radu, et al. "Logical credal networks." Advances in Neural Information Processing Systems 35 (2022): 15325-15337.
>
>
>
> **W4**: Thanks for your advice! We have adapt the suggested models to our datasets with same sample size as in Table 1 and Table 2. We use the metric of success rate (SR) as the key comparison metric. For these models, we hide the event time point information but retain the ordering of events and event type information. Then we can use these models for predicting next action tasks. Given that the Thought Cloning model [6] relies on human thought data (which our model does not require), we assume that the thought event occurs immediately before each action.
>
> | Dataset | Syn Data-1 | Syn Data-2 | Hand-Me-That | Car-Following | MultiTHUMOS | EPIC-Kitchen |
> |---|---|---|---|---|---|---|
> | Thought Cloning [6] | 46.12% | 38.85% | 21.29% | 52.35% | 32.47% | 48.27% |
> | Rule-based ToM [7] | 53.50% | 50.67% | 25.33% | 62.74% | 41.26% | 55.59% |
> | FISER [8] | 56.25% | 51.72% | 32.50% | 66.67% | 40.25% | 61.28% |
> | Ours | 58.28% | 53.15% | 24.72% | 67.28% | 42.80% | 59.74% |
>
> In our synthetic dataset experiments, our model outperforms all baselines in terms of success rate. Due to the absence of human thought data for fair comparison, the thought cloning model performs less effectively on all real datasets compared to other models. For the Hand-Me-That dataset, FISER achieved the highest success rate, with our model and the Rule-based ToM model achieving similar success rates. On the Car-Following and Epic-Kitchen datasets, our model achieves success rates close to FISER and outperforms other models. Specifically, on the Car-Following dataset, our model achieves a success rate of 67.28%, higher than FISHER's 66.67%. On the Epic-Kitchen dataset, our model achieves a success rate of 59.74% close to FISHER's 61.28%. Additionally, on the MultiTHUMOS dataset, our model attains the highest success rate.
>
> Encouragingly, our model has demonstrated competitive performance on both synthetic and real-world datasets. Moreover, it has an additional advantage of predicting future event times, including the timing of goal achievement in the baseline scenarios we examined, a capability absent in other models. Furthermore, in addition to Rule-based ToM, the other two models also lack interpretability compared to our rule-based predictions. Our model can effectively enhance our understanding of the underlying reasons for predictions through logic rules.
>
> [6] Hu, S., & Clune, J. (2024). Thought cloning: Learning to think while acting by imitating human thinking. Advances in Neural Information Processing Systems, 36.\
> [7] Cao, C., Fu, Y., Xu, S., Zhang, R., & Li, S. (2024). Enhancing Human-AI Collaboration Through Logic-Guided Reasoning. In The Twelfth International Conference on Learning Representations.\
> [8] Wan, Y., Wu, Y., Wang, Y., Mao, J., & Jaques, N. (2024). Infer Human's Intentions Before Following Natural Language Instructions. arXiv preprint arXiv:2409.18073.

---

> > ### Author Response · Authors · 2024-11-22
> > **Response to Reviewer 1Fue**
> >
> > **W5**: Your comments are constructive and insightful! In a more complex long-term scenario, subgoals undoubtedly influence the current mental states, consequently impacting actions. Our model operates at a more granular level, focusing on individual subgoal cases. Within each subgoal, we rely on a set of predefined logic rule templates to infer latent mental processes. For long-term goal considerations, an outer layer incorporating a macroscopic router would be necessary to adjust subgoal sequencing based on the current model. This hierarchical model triggers corresponding subset rules for each subgoal, governing distinct mental states in a cascading manner that influences subsequent actions. While our current emphasis lies on a micro-level model in our paper, extending it to long-term horizons is a straightforward extension.
> >
> > In response to your questions:
> >
> > **Q1**: The issue you raised regarding the high computational costs associated with the column generation algorithm is indeed valid. In our model, the presence of prior knowledge, coupled with constraints on rule length during the rule generation and modification processes, mitigates the problem of an overly extensive search space to some extent.
> >
> > Due to the sparse nature of mental events, in most cases, the backtracking mechanism does not trigger a resampling function. The primary motivation behind incorporating the backtracking mechanism is to prevent the omission of these crucial yet sparse mental events. Therefore, the introduction of a backtracking mechanism will not excessively deplete computational resources.
> >
> > To further test the computational cost, we have added more experiments to assess the effect of the backtracking mechanism in real-time applications. We take experiments on Syn Data-1 with 2000 samples for an example:
> >
> > | # of Backtracking Rounds  |  1 |   |   | 2 |   |   | 3 |   |   | 4 |   |   |
> > |---|---|---|---|---|---|---|---|---|---|---|---|---|
> > | # of Rule Templates | 0 | 1 | 2  | 0 | 1 | 2 | 0 | 1 | 2 | 0 | 1 |  2 |
> > | Negative Loss | -24.12  | -22.72  | -20.73  | -21.74  | -20.47  | -18.84  | -19.24  |  -18.43 |  -17.68 | -18.82  | -18.20  |-17.75|
> > | Time Cost (h) | 4.24  | 3.73  | 3.21  | 5.12  | 4.74 | 4.35 | 6.24  | 5.36  |  4.87 |7.00 |  6.51 | 5.89  |
> >
> > Our current selection of 3 rounds of backtracking and 2 provided rule templates is a result of balancing model efficiency and accuracy. Keeping the number of backtracking rounds constant, a decrease in the number of rule templates leads to a drop in the model's converged negative loss and an increase in training time. Conversely, maintaining the same number of rule templates, increasing the rounds of backtracking enhances the model's converged negative loss but also extends training time.
> >
> > Specifically, under our current setup, increasing the rounds of backtracking does not significantly boost the converged negative loss but extends training time to 5.89 hours. Decreasing the rounds of backtracking to 2 results in a drop in the converged negative loss to -18.84. Further reducing the number of provided rule templates to 1 leads to a further decline in negative loss to -20.47, an outcome we aim to avoid.
> >
> > Overall, increasing the backtracking rounds or reducing the provided rule templates did not result in a significant increase in computational cost. Even in the most extreme scenario where backtracking is set to 4 rounds and all rules need to be learned from a blank template (0 prior rule templates provided), the model's convergence time remains at a reasonable 7.00 hours within the current computational infrastructure.

---

> > > ### Author Response · Authors · 2024-11-22
> > > **Response to Reviewer 1Fue**
> > >
> > > **Q2**: We have added more experiments to study the impact of the sample size of the Monte Carlo term on both model performance and computational cost on Syn Data-1.
> > >
> > > | # of MC samples  | 10  | 20 | 30 | 40 | 50 |
> > > |---|---|---|---|---|---|
> > > | Negative Loss  | -19.20 | -19.46 | -17.68 | -17.72 | -17.45 |
> > > | Time Cost (h)  | 3.27 | 3.74 | 4.87 | 5.63 | 6.80 |
> > >
> > > The results suggest that with our current Monte Carlo sample size selection of 30, an increase in sample size leads to a corresponding rise in time cost, although the improvement in negative loss is not significant. However, reducing it to 20 results in a sharp decrease in the model's converged negative loss from -17.68 to -19.46. We prioritize the performance of negative loss over slight improvements in training efficiency. For all the experiments of the paper, our selection of MC sample size is guided by a consistent strategy: to maximize the converged negative loss while maintaining a relatively low training time cost.
> > >
> > > **Q3**: The definition of importance weights of logic rules are aligned with the initial works of temporal logic point process [9, 10].  One can think of the rule weight as the confidence level on the rule. The higher the weight, the more influence the rule has on the intensity. The introduction of rule weights converts the temporal logic rules from "hard" constraints to "soft" constraints. Our point process model uses a weighted combination of temporal logic rules to construct the intensities of the events, which "softened" these logic rules. The probabilistic model allows us to deal with the uncertainty of these irregularly happened events. For our synthetic dataset, the rule weights are predefined. These rule weights can be adjusted as you want when generating new datasets, as long as the updated weights can offer sufficient signals (i.e., higher enough rule weights).
> > >
> > > [9] Li, S., Wang, L., Zhang, R., Chang, X., Liu, X., Xie, Y., ... & Song, L. (2020, November). Temporal logic point processes. In International Conference on Machine Learning (pp. 5990-6000). PMLR.\
> > > [10] Li, S., Feng, M., Wang, L., Essofi, A., Cao, Y., Yan, J., & Song, L. (2021, October). Explaining point processes by learning interpretable temporal logic rules. In International Conference on Learning Representations.
> > >
> > > **Q4**: As we assume both actions and mental processes are temporal point processes, our paper aims to infer latent mental events. The observed action sequences consist of actions that occur in continuous time. For the latent mental processes, we utilize the midpoint or endpoint of the discretized time grid to represent the sampled mental events that occur in continuous time. This does not result in inconsistencies in time scales because both the observed action sequence and the sampled latent mental sequences are now in the same form of point processes, representing sequences of event occurrence times.
> > >
> > > The reason we represent latent mental processes as discrete-time renewal processes is to address the sporadic nature of mental processes and facilitate subsequent inference. It enables us to approximate the intensity function of the latent mental process by stitching together conditional hazard functions. As the discrete interval approaches infinitesimal, it approximates the continuous conditional hazard rate function, thus approximating the continuous conditional intensity function of the mental process.
> > >
> > > **Q5**: Please refer to our response to **W5**.
> > >
> > > **Q6**: In our proposed model, the temporal point processes involve triggering between latent mental and action events, where historical actions can influence latent events and vice versa. Therefore, we resort to the attention mechanism to map the information of the entire action sequence on the spanning time horizon on each discrete time grid. The attention in our model cannot capture how the mental state influences actions. The influence of mental process on action process is reflected on the intensity function after the inference of latent mental process. We have provided an example for visualization of attention weights for the action sequence of Syn Data-1 on discrete time grids as shown in Appendix.L in the revised paper.
> > >
> > > From the figures, we can see that each attention head employs a different pattern to capture dependencies. For each attention head, the impact of one action event is different on each discrete time grid, reflected by various attention weights. The impact of the entire action sequence on a discrete time grid can be conceptualized as a weighted combination of the entire action sequence, with weights derived through attention mechanisms.

---

> > > > ### Comment · Reviewer_1Fue · 2024-11-26
> > > >
> > > > Thank you for your reply. I decide to increase my score slightly.

---

> ### Author Response · Authors · 2024-11-26
>
> Dear Reviewer 1Fue,
>
> We would like to express our gratitude once more for the constructive feedback and prompt response to this paper. We will also ensure the extra results and discussions will be updated in the revised manuscripts.
>
> Warm regards,
>
> The Authors

---

### Official Review · Reviewer_gQch · 2024-11-02

**Soundness:** 4
**Presentation:** 3
**Contribution:** 3
**Rating:** 6
**Confidence:** 4

**Summary:**

The article proposes an encoder-decoder model for inferring human mental states, based on irregular behavioral events to predict future actions. By leveraging temporal point processes and a backtracking mechanism, it captures behavioral dynamics, and introduces logical rules to enhance inference capabilities when data is limited. Experimental results demonstrate that this model outperforms baseline methods in intention inference and prediction accuracy.

**Strengths:**

The formulation in the article is relatively clear in terms of definitions and inference logic, particularly in the sections on conditional entropy and probabilistic modeling (such as the conditional entropy expression in formula (15)). These formulas effectively capture the temporal relationships between events.
The experiments in the article are relatively thorough, considering a comprehensive range of comparisons.

**Weaknesses:**

1. The model's performance is dependent on the quality of the logical rules defined and generated; if the rules are of low quality, it may affect inference results.
2. The backtracking mechanism may increase computational complexity, and further discussion on its design could be interesting. Consider considering a priority filtering mechanism within the model to conduct backtracking checks only on the most relevant events, which could make the approach more intriguing.
3. Currently, the model relies on predefined logical rule templates. While this offers strong initial inference capabilities, it may limit the model's adaptability in complex scenarios.

**Questions:**

1. If the prediction remains inaccurate after multiple rounds of backtracking, how does the mechanism adjust or avoid getting stuck in an infinite loop?
2. The paper mentions that the rule generator uses predefined templates and a column generation algorithm to generate logic rules, but the specific process and optimization details are not described. How have the most critical rules for improving model performance been identified?
3. Logic rules are embedded as prior knowledge to enhance inference capabilities, but do different datasets have different requirements for rules?
4. Does the performance of the backtracking mechanism vary depending on the embedded logic rules?

---

> ### Author Response · Authors · 2024-11-22
> **Response to Reviewer gQch**
>
> We are grateful for your careful review! We will address your concerns from the following aspects.
>
> In response to your mentioned weaknesses:
>
> **W1 & W3**: Although in our current work we incorporate rule templates as priors to harness domain knowledge, our rule-generation algorithm is inherently capable of learning logic rules from scratch, even in scenarios with limited prior knowledge. Utilizing the column-generation framework, the algorithm autonomously uncovers and refines new rules, with its accuracy improving as more data becomes available. Additionally, standard techniques such as constraining rule lengths ensure computational efficiency and prevent an excessively large search space.
>
> In practice, we can employ two strategies tailored to distinct scenarios: autonomous rule learning for data-rich domains and template-guided learning for scenarios with limited data but ample prior knowledge.
>
> In our response to **Reviewer orep, W7**, we evaluate the effectiveness of our model in scenarios with and without predefined rules, across different sample sizes. Please refer to the table for detailed experiment results.
>
> The results demonstrate that under the same training sample cases, not providing prior rule templates during training leads to basically satisfactory prediction accuracy and convergence negative loss, but increased training time. When increasing the training sample size without prior rule templates, the model exhibits a notable improvement in converged negative loss and prediction accuracy, approaching or even matching the model performance achieved with prior rules on a smaller dataset. Therefore, with ample data, our model can deliver strong performance even without prior knowledge.
>
> If the quality or quantity of predefined rules is low, the presence of the backtracking mechanism can also help improve model performance. In our response to **Review 1Fue, Q1**, we have added more experiments to assess the effect of the backtracking mechanism and predefined rule templates. The results indicate that reducing the quantity of predefined rules, coupled with appropriately configured backtracking rounds, can still yield a higher converged negative loss and prediction accuracy comparable to cases with more predefined rule templates and less backtracking rounds.
>
> **W2 & Q4**: Thanks for your advice! Adding a priority filtering mechanism would be a promising approach to improve the efficiency of the backtracking mechanism. We can establish predefined filtering mechanisms based on logic rules to selectively discard certain events during the backtracking process with tunable probability thresholds. This approach may enhance efficiency while maintaining exploration capabilities.
>
> During the rebuttal phase, we have incorporated ablation studies to evaluate how the number of backtracking rounds and the prior logic rule templates affect the model performance and computational efficiency. Please refer to our reply to **Review 1Fue, Q1** for the detailed experimental results.
>
> Due to the sparse nature of mental events, in most cases, the backtracking mechanism does not trigger a resampling process. Therefore, the introduction of a backtracking mechanism will not excessively deplete computational resources.  The experimental results also confirm this point, by setting a reasonable number of backtracking rounds, we enhance model performance with only a slight increase in computational time. Additionally, increasing the number of backtracking rounds does not excessively raise computational cost.
>
> Considering the impact of prior rule templates, when holding the backtracking rounds unchanged, reducing the provided prior rule templates can somewhat diminish model performance. However, under the same set of provided prior rules, such as offering only one prior rule, increasing the backtracking rounds from 1 to 3 elevates the converging negative loss from -22.72 to -18.43. Therefore, the results demonstrate that the presence of the backtracking mechanism helps mitigate the decrease in model performance caused by the lack of well-defined prior rules.

---

> ### Author Response · Authors · 2024-11-22
> **Response to Reviewer gQch**
>
> In response to your questions:
>
> **Q1**: Firstly, we want to clarify that the probabilities of our model's latent mental processes and the intensities of the action processes are non-negative. Considering the resolution of the discretized time grid is not infinitesimal, theoretically, infinite loops should not occur.  Moreover, we have considered multiple possible ways to improve the accuracy of the backtracking mechanism:
>  - **Logic-based Event Selection**: we can filter some events based on the learned or predefined logic rules.
>  - **Learn from Failures**: we can maintain a history of failed sampled events to avoid making the same mistakes again.
> - **Conflict-Directed Backtracking**: we can backtracking more than one level if inference and prediction are not stable.
>
> **Q2**: In Appendix B, we have provided details of our rule generator. For the rule generation process, we first extract temporal logic rule templates from prior knowledge, such as expert knowledge. We can also learn temporal logic rules starting from a blank template. Then we use the column generation algorithm to generate temporal logic rules. The algorithm alternates between the rule generation stage and the rule evaluation stage [1]. In the rule evaluation stage, we solve the restricted master problem which evaluates the current rules (updates rule base and rule weights) by maximizing the likelihood:
>
> $$\text{Restricted Master Problem}: \mu^*_k, w^*_k = argmin -l(\mu, w) + \Omega(w),\ s.t.,\ w_f \geq 0, f \in F_k$$
>
> where $F_k$ is the current rule set, $l(\mu, w)$ is the log-likelihood derived by Equation (3) in our paper, and $\Omega(w)$ is a convex regularization function on the rule length.
>
> In the rule generation stage, we solve the subproblem to search and construct a new temporal logic rule based on the given template (or blank template).
>
> $$\text{Sub Problem}: min_{\phi_f} -\frac{\partial l(\mu, w)}{w_f} + \frac{\Omega(w)}{w_f} |_{\mu^*_k, w^*_k}$$
>
> The rule generator module act as a plug-in module, if prior rule templates were given, we can consider adding and removing predicates and corresponding temporal relations to refine the prior knowledge in sub-problem. Or freeze the rule generator module and solely rely on the prior knowledge
> to guide the generating process. If prior rule templates were not given, we can still use the rule generator to search the rule from scratch.
>
> [1] Li S, Feng M, Wang L, Essofi A, Cao Y, Yan J, & Song L (2021, October). Explaining point processes by learning interpretable temporal logic rules. In International Conference on Learning Representations
>
> **Q3**: For different situations or datasets, our proposed model needs to specify different logic rule templates for different datasets which should be easily obtained from common sense, expert knowledge, government regulations, and literature references. Simultaneously, in cases where we lack sufficient prior knowledge, our model can also learn rules from blank templates, without the need of predefined rules.
>
> In high-stakes scenarios, it is common that many existing works predefine various logic rules as prior knowledge across different domains to boost model performance. For instance, in [2], traffic laws were formalized using linear temporal logic based on government regulations and expert knowledge. In [3], domain knowledge summarized by human experts in the form of first-order logic rules was incorporated to model event dynamics via intensity function. The proposed model in [4] encoded prior knowledge as temporal logic and power function, facilitating the learning of neural point process models for inferring coordinated behaviors in social media. Additionally, [5] utilized imprecise expert knowledge in first-order logic formulas to specify a set of probability distributions over all its interpretations. Similar to these work, our proposed model also incorporates prior knowledge in first-order logic form for different scenarios.
>
> [2] Liu, Jiaxin, et al. "Semantic traffic law adaptive decision-making for self-driving vehicles." IEEE Transactions on Intelligent Transportation Systems (2023).\
> [3] Li, Shuang, et al. "Temporal logic point processes." International Conference on Machine Learning. PMLR, 2020.\
> [4] Zhang, Yizhou, Karishma Sharma, and Yan Liu. "Vigdet: Knowledge informed neural temporal point process for coordination detection on social media." Advances in Neural Information Processing Systems 34 (2021): 3218-3231.\
> [5] Marinescu, Radu, et al. "Logical credal networks." Advances in Neural Information Processing Systems 35 (2022): 15325-15337.

---

> ### Comment · Reviewer_gQch · 2024-11-25
>
> Thank you for your reply and I will keep my scores.

---

> ### Author Response · Authors · 2024-11-26
>
> Dear Reviewer gQch,
>
> We deeply appreciate your positive comments and valuable feedback. We will also ensure to incorporate the improvements in the final version.
>
> Warm regards,
>
> The Authors

---

### Official Review · Reviewer_4kqs · 2024-11-04

**Soundness:** 2
**Presentation:** 2
**Contribution:** 3
**Rating:** 5
**Confidence:** 2

**Summary:**

The paper introduces an encoder-decoder framework that infers latent human mental states (e.g., beliefs, intentions) from irregularly observed action events and predicts future actions. By integrating a temporal point process model with logic rules as priors and a backtracking mechanism, the model improves prediction accuracy and adapts effectively to real-time changes. Its performance is validated through experiments on synthetic and real-world datasets, outperforming existing baselines.

**Strengths:**

- The research problem is valuable in perception level task and can contribute to practical applications such as autonomous driving
- Thorough experiments reveal promising results

**Weaknesses:**

- Limited readability: the paper missed several key points for readers without much related background to understand the task. For example, is the task generative or predictive (i.e., does it generate the action and mental thoughts directly or does it select from an action and mental space); in a high level, how does the backtracking mechanism decide whether to generate/predict a mental state or an action? Instead of explaining a specific step in the mechanism, readers may want to understand the mechanism from the motivation.
- If the model is simply selecting a mental state or an action out of a candidate pool, then the model is not able to intrinsically understand the environment and advanced human perception. Generating and mimicking human behaviors requires the model to process the information from the environment and act like human based on word knowledge and rules, which the proposed framework cannot comprehend as it has not seen it all (i.e., gone through massive pre-training). Therefore, I challenge the "understand" behavior referred in the title.

**Questions:**

See weaknesses.

---

> ### Author Response · Authors · 2024-11-22
> **Response to Reviewer 4kqs**
>
> We thank reviewer 4kqs for the time of reviewing and the constructive comments! We will address your concerns below.
>
> **W1: Paper Clarity**:  We apologize for any unclarity in our paper. In the revised version, we have included more in-depth insights into the underlying mechanisms, motivations, and interconnections among the various modules of the model. The updated paper has been re-uploaded.
>
> **W2: Generative v.s. Predictive**: First, generative models are highly effective for predictive tasks, often complementing predictive approaches, particularly in sequential settings (e.g., deep autoregressive models, which are generative models, used for predicting future sequences).
>
> Our model employs a **generative approach** for both actions and mental states, serving distinct purposes. For actions, the generative process **predicts** future events by sampling from their conditional probability distribution, enabling accurate forecasting of behavior. For mental states, the generative approach facilitates learning and inference by sampling latent trajectories from an approximate posterior distribution, as the true posterior lacks a closed form. These sampled trajectories allow the model to marginalize over the posterior effectively, supporting both the understanding of latent cognitive processes and the prediction of observable actions.
>
> **W3: Insight for Backtracking Mechanism**: In a nutshell, the primary motivation behind incorporating the backtracking mechanism is to prevent the omission of the crucial yet sparse mental events. In real-life scenarios, human behavior evolves with changing thoughts. Considering the example mentioned in Section 3 _Discrete-Time Renewal Process_ in our paper, if someone is thinking about starting an exercise routine, typically, they are likely to promptly begin exercising. However, if they experience a shift in mindset, such as feeling tired, they might struggle to maintain a consistent exercise routine. Therefore, these shifts in thinking are typical and should be noted, as humans frequently ponder prior thoughts before making definitive decisions.
>
> We introduce a backtracking mechanism based on this **human cognitive and behavior logic**. When sampling actions, it's hard to ensure that new thoughts won't arise between consecutive actions. Hence, the backtracking mechanism is utilized to examine if new thoughts have emerged during this time interval. If so, these are considered and actions are resampled. Through iterative backtracking, sampled actions can be refined continuously and newly emerged thoughts are not overlooked, enhancing adaptability to real-time fluctuations in human behavior.
>
> **W4: Generating from the Candidate Pool vs. Novel Action/Mental Events**: In our framework, actions and mental states are generated within a predefined candidate pool, reflecting a closed problem where all possible actions and mental events are specified in advance.
>
> While our current approach is grounded in a closed setting, the framework could be extended to open-ended problems by introducing _predicate invention mechanisms_ for generating novel mental states. We agree with the reviewer that such an extension would make the model more flexible and more capable of handling dynamic, real-world scenarios.
>
> **W5: Our claim about _understanding_**: Even within the closed problem setting, we assert that our framework can still **understand** human behavior. This understanding is achieved in two significant ways:
> - **Learning Rule Content**: The model uncovers the underlying structure of human decision-making by learning the content of behavioral rules—how actions are generated and how they depend on inferred mental states.
> - **Personalized Rule Usage**:  Beyond the rules themselves, the model also learns individual preferences in applying these rules. By modeling the conditional probability of actions and mental states, the framework understands the evolution of human thoughts and captures how different humans prioritize and weigh various rules, reflecting personalized behavioral patterns.

---

> > ### Comment · Reviewer_4kqs · 2024-11-26
> >
> > I appreciate the authors for the responses. I have carefully reviewed the responses and decide to keep my score. Thanks.

---

> ### Author Response · Authors · 2024-11-27
> **Sincerely expecting further discussion from Reviewer 4kqs**
>
> Dear Reviewer 4kqs,
>
> We thank reviewer 4kqs for the prompt response to this paper. We wonder whether our response has addressed your concerns.
>
> In our response, we have:
>
> (1) improved the writing of our paper and uploaded the revised version.\
> (2) clarified the distinct roles of generative process in predicting action events and inferring latent mental events.\
> (3) detailed the high-level motivation behind the proposed backtracking mechanism and highlighted its alignment with human cognitive and behavioral logic.\
> (4) clarified that our framework understands human behavior by learning rule content, modeling probability of actions and mental states, and capturing personalized rule usage.
>
> We sincerely inquire if reviewer 4kqs has any additional concerns, and we are more than willing to provide corresponding response. Thank you!
>
>
> Best regards,
>
> The Authors

---

### Official Review · Reviewer_orep · 2024-11-05

**Soundness:** 2
**Presentation:** 1
**Contribution:** 2
**Rating:** 5
**Confidence:** 2

**Summary:**

This paper proposed a model combining rule-based learning approaches with statistical approaches to infer evolving mental states and the resulting action seuquences of human beings. The proposed model is advantageous to predict some irregular action events, which is the severest weakness in statistical approaches that strongly depends on sufficient data collections. The proposed decoder is constituted of ruled-based modules which can provide interpretability to some extent. Furthermore, the proposed backtracking mechanism can detect the hiddenly evolving mental states and thus correct the action predictions. The experimental results demonstrate the effectiveness of the proposed approach.

**Strengths:**

1. This paper combined the statistical model with the rule-base learning approaches, in addressing human behavior prediction. Although in general this kind of combination has been always a research trend in rule-based learning comminity, it is still an intriguing and original approach to be applied in this research problem. One of prominent reasons is that the incorporation of rules can indeed give some interpretation of human behaviors prediction, especially when data is lacking. In my view, this is no doubt a trend of machine learning research in the future.
2. The description of the proposed approach is detailed, with sufficient descriptions on how each module is modelled. The experimental settings are clear and results are compred with multiple baseline algorithms. The illustration of figures helps understand the virtue of the proposed approach.
3. Although this paper lacks some explanations about why some modules are modelled in specific ways, the general description of the proposed modules have been clearly explained. Also, it has evident motivations and sufficient experimental results. Standing on all these point, the quality of this paper is moderate.
4. The research question studied in this paper is significant. Human behavior prediction is always a fundation in dowstream tasks such as human-Al interaction. Most of existing reserch focused on employing pure statistical models to answer this question. One reason is that statistical machine learning has stood out in contrast to logic-based systems since 1990s, and thus people who are knowledgable in logics are not as many as people who are expertised in statistical machine learning. From this perspective, the incorporation of (temporal) logic-based learning into statistical machine learning implemented in this paper is worth encouraging.

**Weaknesses:**

1. The paper writing is not good enough. Except that the motviation of this paper (research question) is easy to understand, the rest of contents are more like accumulation of descriptions, lacking clear logic to link the contituents (paragraphs) of this paper.
2. Typo in line 158-159: Defome -> Deform
3. In line 170-171, you mentioned that the instantaneous event probability is modelled via the hazard function. Could you give more reasons to explain why the hazard function is a the choice? For example, what is the advantage of this model than others?
4. In line 179, you mentioned that once a mental event occurs, the discrete-time survival process resets. How is this statement justified by equation (1) and equation (2)? Some more details are needed.
5. Why is the time embedding for the observed action time modelled as equation (4)?
6. Why is the probabilistic model to sample mental event type modelled in equation (8)? Especially, is there any insights into adding log probability (\\(p_\{|xi\}|)) to collected samples (from \\(g_\{|xi\}\\))?
7. One of the most serious weaknesses to logic-based learning is the construction of logic rule templates. I would like to see some more discussion on the remedy for dealing with situations that are less knowledgable or strange to humans.
8. It is evident that the resolution of discretized time grid would strongly influence the performance of the proposed approach. Could you give some more discussion on some potential strategy to decide this hyperparameter?

**Questions:**

Please address t he concerns in weaknesses.

---

> ### Author Response · Authors · 2024-11-22
> **Response to Reviewer orep**
>
> We sincerely thank you for your insightful reviews! We hope our response below addresses your concerns.
>
> **W1**: We apologize for any ambiguity in our paper. In the revised version, we have improved clarity by offering deeper insights into the proposed model and enhancing the logical flow between sections. The revised version has been uploaded.
>
> **W2**: Thank you for pointing it out! We have revised the sentence to "Define an action set…" in the updated version.
>
> **W3**: We use the hazard function to model the instantaneous event probability because it effectively captures dynamic event likelihoods **over time**.
>
> By definition, the hazard $h(t):=\frac{p(t)}{S(t)}$, where $p(t)$ represents the probability of an event occurring at the specific time $t$, and $S(t)$ is the survival function, indicating the probability that no event has occurred up to time $t$.
>
> The reason we use the hazard function to model event rates is that it effectively captures the evolution of action or mental states based on the _**elapsed time** since the last event_. Moreover, the hazard function can be adapted by incorporating historical or external features, enhancing the model's flexibility to account for varying dynamics.
>
> We use a **discrete-time hazard function** to model the occurrence probability of hidden mental events and a continuous-time hazard function to model the occurrence rate of action events.
>
> - **Justification for discrete-time hazard function**: By discretizing time and utilizing the renewal process framework, we gain **computational tractability**, making it feasible to infer latent mental states over time.
> - **Justification for continuous-time hazard function**: The continuous-time hazard function allows us to model real-time event predictions for action events, accommodating their continuous and dynamic nature.
>
> This dual approach ensures an effective and computationally efficient model for both mental and action events. We have added this detailed explanation in our revised paper, please refer to Section 3.
>
> **W4**: The resetting of the survival process after an event is intrinsic to the renewal process definition. According to renewal property: In a renewal process, the occurrence of an event marks the end of one cycle and the start of another. As defined in Equation (1), the survival function $S\left(t_{\xi}\right)$, which represents the probability that no mental event has occurred up to time $t_{\xi}$, is reset to 1 immediately after an event.
>
> Equation (2) defines the event probability $p_{\xi}=S\left(t_{\xi-1}\right)-S\left(t_{\xi}\right)$, which approximates the likelihood of an event occurring in the interval $V_{\xi}$. After an event, $S\left(t_{\xi}\right)$ resets to 1 at the next interval, ensuring that both the survival and event probabilities reflect the reset process. We have also added the corresponding analysis in Section 3 in our revised paper.
>
> **W5**: We use the embeddings for the observed action time as defined in Equation (4) to capture the information of the action sequence on the spanning time horizon and address irregularities. The embedding  method we used in our paper is commonly employed in attention mechanisms for processing sequential event data [1, 2]. This temporal encoding preserves sequence order, encodes temporal information, handles irregular time gaps, aids the attention mechanism in capturing long-range dependencies between events, and reduces symmetry in self-attention mechanisms.
>
> [1] Zuo, S., Jiang, H., Li, Z., Zhao, T., & Zha, H. (2020, November). Transformer hawkes process. In International conference on machine learning (pp. 11692-11702). PMLR. \
> [2] Yang, C., Mei, H., & Eisner, J. (2021). Transformer embeddings of irregularly spaced events and their participants. arXiv preprint arXiv:2201.00044.
>
> **W6**: We sample the mental event type according to Equation (8) using the Gumbel-Softmax trick [3], which is a method used to approximate the sampling of discrete variables with a continuous relaxation, enabling the backpropagation of gradients through the sampling process. This trick ensures end-to-end differentiable during the training of neural networks involving discrete outputs, such as when sampling discrete mental event types in our scenario. The reason for using log probabilities is to ensure numerical stability. In mathematically form, this trick can be written as
> $$y_i = \frac{\exp((\log(\pi_i) + g_i) / \tau)}{\sum_{j=1}^k\exp((\log(\pi_j) + g_j)/\tau)}, \text{for}\ i=1,...,k$$
> where $g_i$ is the Gumbel noise, $\pi_i$​ is the probability of category $i$, and $\tau$ is the temperature parameter that controls the smoothness of the softmax. Our Equation (8) is based on this initial formula.
>
> [3] Jang, E., Gu, S., & Poole, B. (2016). Categorical reparameterization with gumbel-softmax. arXiv preprint arXiv:1611.01144.

---

> ### Author Response · Authors · 2024-11-22
> **Response to Reviewer orep**
>
> **W7**: Although in our current work we incorporate rule templates as priors to leverage the rich domain knowledge of the application, our proposed rule-generation algorithm is inherently capable of learning logic rules from scratch, even in scenarios with limited prior knowledge. By leveraging the column-generation framework, the algorithm can autonomously discover and refine new rules, with its accuracy improving as more data becomes available. Additionally, standard techniques such as constraining rule lengths ensure computational efficiency and prevent an excessively large search space.
>
> In practice, we can employ two complementary strategies tailored to different problem settings: autonomous rule learning for data-rich domains and template-guided learning for scenarios with limited data but abundant prior knowledge.
>
> To assess our model's efficacy in scenarios without predefined rules, we have conducted an ablation study in Appendix I to evaluate its performance without prior logic rule templates. Below, we present expanded results on Syn Data-1 across various sample sizes.
>
> |Sample Size   | 2000  |   | 5000  |   |
> |---|---|---|---|---|
> | Case | Prior Rules -- No  | Prior Rules -- Yes  | Prior Rules -- No | Prior Rules -- Yes |
> | Negative Loss | -19.24 | -17.68 | -18.25 | -17.20 |
> | ER% | 43.76% | 41.72% | 42.33% | 41.25% |
> | MAE | 2.58 | 2.32 | 2.39 | 2.28 |
> | Time Cost (h) | 6.24 | 4.87 | 8.36 | 6.92 |
>
> With 2000 training samples and no prior rule templates provided, there was a minor accuracy decrease in the prediction task, but the model still achieved an ER% of 43.76% and an MAE of 2.58. Increasing the training samples to 5000 without prior rule templates led to an improvement in the model's negative loss at convergence from -19.24 to -18.25. The model's performance in the prediction task also improved, with the ER% and MAE reaching 42.33% and 2.39, respectively. Encouragingly, these results closely approach those achieved with 2000 samples and provided prior rule templates. These results showcase that our model provides viable strategies for handling scenarios where prior knowledge is lacking. Furthermore, with sufficient data, our model can achieve good performance even in the absence of prior knowledge.
>
> For scenarios with limited data but ample prior knowledge, we can employ an alternative strategy to integrate prior knowledge. In Appendix.H, the results of the scalability experiment demonstrate that our model can handle limited data issues. For instance, with prior rule templates provided, reducing the sample size from 2000 to 1000 resulted in only a slight increase of 2.25% in ER% and 0.19 in MAE for Syn Data-3, and a 2.36% increase in ER% and 0.25 in MAE for Syn Data-4 (Syn Data-3 and Syn Data-4 are datasets with more complex ground truth rules and larger domains for latent mental states). These results demonstrate the robustness of our model with small datasets.
>
> Overall, We can appropriately select remedies for providing prior knowledge to the model based on varying dataset sizes and levels of prior knowledge.

---

> ### Author Response · Authors · 2024-11-22
> **Response to Reviewer orep**
>
> **W8**: Thanks for your suggestion! In our experiment, we use **metrics** such as training negative loss, training time, and prediction accuracy to select appropriate resolution of a discretized time grid. The choice of resolution balancing accuracy and sample efficiency. For example, in experiment on Syn Data-1, we use grid search for the resolution and the results are as follows:
>
> | Resolution  | 0.25 | 0.50 | 0.75 | 1.00 | 1.25 |
> |---|---|---|---|---|---|
> | Negative Loss | -16.90 | -17.68 | -18.24 | -19.43 | -20.74 |
> | ER% | 40.67% | 41.72% | 42.74% | 45.72% | 45.90% |
> | MAE | 2.28 | 2.32 | 2.31 | 2.53 | 2.62 |
> | Time Cost (h) | 6.57 | 4.87 | 4.41 | 3.85 | 3.69 |
>
> From the results one can see that under current selection of 0.50, the proposed model achieves the second highest negative loss. Reducing the resolution to 0.25 only marginally increases the negative loss to -16.90, but extends the model convergence time by 1.70 hours. Conversely, while increasing the resolution can only slightly shorten training time. Moreover, it only slightly increase the model's converged negative loss and result in a decline in prediction accuracy when using the well-trained model for prediction tasks. This could be due to missing some crucial latent mental events. Hence, balancing between accuracy and computational efficiency, we opt for the current resolution of 0.5. This strategy is applied consistently for other datasets to determine the resolution.

---

> ### Author Response · Authors · 2024-11-27
> **Sincerely expecting further discussion from Reviewer orep**
>
> Dear Reviewer orep,
>
> We thank reviewer orep for the time of reviewing and the constructive comments. We really hope to have a further discussion with reviewer orep to see if our response resolves the concerns.
>
> In our response, we have:
>
> (1) enhanced clarity, delved deeper into the model, and improved coherence across sections in our paper. We have also uploaded the revised version. \
> (2) explained the hazard function and reset mechanism in discrete-time survival processes, elucidated the embedding technique for temporal event sequences, and detailed the gumbel-softmax trick. \
> (3) clarified that our model excels in autonomous rule learning for data-rich domains, and template-guided learning for scenarios with limited data but abundant prior knowledge. Additionally, we extended the ablation study to evaluate our model's effectiveness in scenarios lacking prior knowledge, showing its ability to perform well even in the absence of prior knowledge within a reasonable training timeframe. \
> (4) clarified that we determined the resolution of the discretized time grid based on metrics like training loss, time, and prediction accuracy, supported by corresponding experimental results.
>
> We genuinely hope reviewer orep could kindly check our response. If reviewer orep has any further questions, we are more than willing to response. Thanks!
>
> Best regards,
>
> The Authors

---

> ### Author Response · Authors · 2024-12-01
> **Invitation for review feedback**
>
> Dear Reviewer orep,
>
> Thank you very much for your commitment and effort in evaluating our paper. We understand that this is a hectic time, and we are writing to gently remind you about providing your insights on our rebuttal. As the discussion phase is drawing to a close, your feedback would be invaluable to us.
>
> Should you have any further thoughts or recommendations about our work, we are eager to discuss them with you.
>
> We eagerly await your reply.
>
> Thank you once again,
>
> The Authors

---

> ### Author Response · Authors · 2024-12-03
>
> Dear Esteemed Reviewer orep,
>
> We are grateful to you for generously dedicating time and effort to evaluating our paper. As the deadline of reviewer-author discussion is fast approaching, we are becoming increasingly concerned about the absence of feedback or response from you.
>
> We fully recognize and appreciate the significant efforts and time constraints faced by reviewers. We have responded to your major concerns about the "efficiency of prior logic rules", "selection of resolution of discretized time grid", and "clarification about hazard function, temporal embedding, and gumbel-softmax trick" point-by-point in the response sections.
>
> Nevertheless, considering the critical role of the rebuttal phase in the review process, we are worried that the lack of interaction with you could potentially affect the fair and thorough assessment of our work. If you have any further concerns, please let us know and we are willing to response!
>
> Warm wishes,
>
> The authors

---

### Author Response · Authors · 2024-12-03

Dear Reviewers, Senior Area Chairs, Area Chairs, and Program Chairs,

We are deeply grateful for the insightful comments and suggestions, which are invaluable for enhancing our work. We are excited that the reviewers hold positive feedback and find our work “The research question studied in this paper is significant. The proposed model has evident motivations and sufficient experimental results” (Reviewer orep), “The research problem is valuable and can contribute to practical applications. The experiments are thorough and reveal promising results” (Reviewer 4kqs), “The formulation is relatively clear, and the experiments are relatively thorough, considering a comprehensive range of comparisons” (Reviewer gQch), and “This paper proposes a novel model for asynchronous action sequence modeling and inferring latent mental events. The integration of rule-based logic for dynamic prediction adjustment presents an interesting angle.” (Reviewer 1Fue).

In our response, we have included additional clarifications and experiments. Recognizing the importance of further explanations and experiments, we are committed to these enhancements, reflected in corresponding modifications and supplements in our revised paper. To ensure transparency and clarity, we have outlined our main responses as follows:

- **Scalability and computational cost**: In response to the Reviewer 1Fue' suggestion, we have expanded our evaluation to include two additional synthetic datasets and two real-world datasets with varying sample sizes. The new synthetic datasets feature complex rules and expanded latent state domains. As demonstrated in Figure 8 of Appendix H in the revised paper, our method exhibits promising predictive performance with small-scale datasets and improves as the sample size increases, converging rapidly even with large-scale datasets.

- **Importance of logic rules**: The introduction of prior logic rules can help to deal with limited-data issue. Moreover, our model can employ two complementary strategies tailored to different problem settings: autonomous rule learning for data-rich domains and template-guided learning for scenarios with limited data but abundant prior knowledge. In our response to Reviewer orep, Reviewer gQch, and Reviewer 1Fue, we have evaluated the effectiveness of our model in scenarios with and without predefined rules, across different sample sizes. Without prior knowledge, our proposed model can still achieve satisfactory performance by learning rules from scratch, showing good ability to adapt to data lacking prior knowledge.

- **Efficiency of backtracking mechanism**: Due to the sparse nature of mental events, in most cases, the backtracking mechanism does not trigger a resampling function. Therefore, the introduction of a backtracking mechanism will not excessively deplete computational resources. In our response to Reviewer gQch and Reviewer 1Fue, we have added more experiments to assess the effect of the backtracking mechanism in real-time applications. Results indicate that increasing the backtracking rounds did not result in a significant increase in computational cost. Our current selection of backtracking rounds is a result of balancing model efficiency and accuracy.

- **Hyper-parameter selection**: We have used metrics such as training negative loss, training time, and prediction accuracy to select hyper-parameters such as resolution of a discretized time grid and number of Monte Carlo samples. Experimental results can be found in our response to Reviewer orep and Reviewer 1Fue.

- **Comparison with extra baselines**: As suggested by Reviewer 1Fue, we compare our model with three more SOTA baselines (Thought cloning, Rule-based ToM, and FISER). Our model has demonstrated competitive performance on both synthetic and real-world datasets. Moreover, our model has additional advantages of predicting future event times and enhancing interpretability by incorporating logic rules.

In addition to the academic contributions, our method holds practical significance. By incorporating prior logic rules, our proposed model tackle with limited data issue and enhance interpretability. By inferring latent mental events and incorporating rule learning, our method enhances AI's ability to understand human thoughts and predict their actions, enabling timely assistance and promoting human-AI collaboration. We believe this innovative approach has the potential to inspire future research endeavors.

---

### Meta-Review · Area_Chair_VHb8 · 2024-12-21

**Metareview:**

This paper presents a versatile encoder-decoder model that infers evolving human mental processes and predicts future actions using a temporal point process framework to address irregular event observations and limited data. By incorporating logic rules as priors and a backtracking mechanism, the model enhances the accuracy of predictions and demonstrates strong performance on both synthetic and real-world datasets, contributing to more human-centric AI systems. Despite its contributions, according to the reviewers' feedback and discussion, this paper still needs to be improved on the following aspects: writing presentation, insufficient discussion on unfamiliar situations, ambiguity in task nature, challenges in true understanding of human behavior, dependence on rule quality, potential computational complexity, and limited adaptability due to predefined templates.

**Additional Comments On Reviewer Discussion:**

The discussions are in multi-turns. The reviewers provided their feedbacks conditioned on the authors rebuttal.

---

### Decision · Program_Chairs · 2025-01-22

Reject